# Pak1 and PP2A antagonize aPKC function to support cortical tension induced by the Crumbs-Yurt complex

Cornelia Biehler[1,2], Katheryn E Rothenberg[3,4], Alexandra Jette[1,2], Helori-Mael Gaude[1,2], Rodrigo Fernandez-Gonzalez[3,4,5,6], Patrick Laprise[1,2]*

[1]Centre de Recherche sur le Cancer, Université Laval, Québec, Canada; [2]axe oncologie du Centre de Recherche du Centre Hospitalier Universitaire de Québec-UL, Québec, Canada; [3]Institute of Biomedical Engineering, University of Toronto, Toronto, Canada; [4]Ted Rogers Centre for Heart Research, University of Toronto, Toronto, Canada; [5]Department of Cell and Systems Biology, University of Toronto, Toronto, Canada; [6]Developmental and Stem Cell Biology Program, The Hospital for Sick Children, Toronto, Canada

**Abstract** The *Drosophila* polarity protein Crumbs is essential for the establishment and growth of the apical domain in epithelial cells. The protein Yurt limits the ability of Crumbs to promote apical membrane growth, thereby defining proper apical/lateral membrane ratio that is crucial for forming and maintaining complex epithelial structures such as tubes or acini. Here, we show that Yurt also increases Myosin-dependent cortical tension downstream of Crumbs. Yurt overexpression thus induces apical constriction in epithelial cells. The kinase aPKC phosphorylates Yurt, thereby dislodging the latter from the apical domain and releasing apical tension. In contrast, the kinase Pak1 promotes Yurt dephosphorylation through activation of the phosphatase PP2A. The Pak1–PP2A module thus opposes aPKC function and supports Yurt-induced apical constriction. Hence, the complex interplay between Yurt, aPKC, Pak1, and PP2A contributes to the functional plasticity of Crumbs. Overall, our data increase our understanding of how proteins sustaining epithelial cell polarization and Myosin-dependent cell contractility interact with one another to control epithelial tissue architecture.

*For correspondence:
Patrick.Laprise@crchudequebec.
ulaval.ca

Competing interests: The authors declare that no competing interests exist.

## Introduction

Simple epithelia form cohesive barriers owing to specialized adherens junctions. For instance, the cadherin–catenin complex, a major component of the *zonula adherens* (ZA), links cortical actin filaments of neighboring cells indirectly to ensure strong cell–cell adhesion (*Harris and Tepass, 2010*). Association of the hexameric motor protein non-muscle myosin II (hereafter, myosin) with ZA-associated actin bundles ensures contractility (*Mège and Ishiyama, 2017*). The ability of epithelial cells to support vectorial transport and secretion adjusts the biochemical environment on both sides of the epithelial layer. These unidirectional functions require the polarization of epithelial cells along the apical–basal axis (*Laprise and Tepass, 2011*). Reciprocal interactions between the epithelial polarity protein network and ZA components not only define the functional architecture of individual epithelial cells, but also shape epithelial tissues (*Tepass, 2012*). A classic example is the formation of specialized epithelial structures, such as tubes or acini, resulting from concerted myosin-dependent apical constriction of a group of cells within epithelial sheets (*Martin and Goldstein, 2014*).

The apically localized protein Crumbs (Crb) acts as an apical determinant in *Drosophila* epithelia (*Pocha and Knust, 2013*; *Tepass, 2012*; *Tepass et al., 1990*; *Wodarz et al., 1995*). The cytoplasmic tail of Crb contains a PSD95/Dlg1/ZO-1 (PDZ)-domain binding motif (PBM), and a Four point one,

Ezrin, Radixin, Moesin (FERM)-domain binding motif (FBM) (*Klebes and Knust, 2000*; *Klose et al., 2013*; *Pocha and Knust, 2013*; *Tepass, 2012*). The PBM recruits PDZ-containing proteins such as Stardust (Sdt), which cooperates with Crb in establishing the apical membrane and promoting its growth (*Bachmann et al., 2001*; *Pocha and Knust, 2013*; *Tepass and Knust, 1993*). The FBM interacts with several FERM domain proteins, including Yurt (Yrt) (*Laprise et al., 2006*; *Tepass, 2009*). The latter is localized to the lateral membrane throughout embryogenesis, and supports the stability of this membrane domain during organogenesis (*Laprise et al., 2006*; *Laprise et al., 2009*). During terminal differentiation of epithelial cells derived from the ectoderm, Yrt also occupies the apical membrane due to a direct interaction with Crb (*Laprise et al., 2006*). The apical recruitment of Yrt limits Crb-dependent apical membrane growth to establish a specific apical/basolateral membrane size ratio, which is crucial to defining the morphometric parameters of developing epithelial structures (*Laprise et al., 2006*; *Laprise et al., 2010*). Crb displays cooperative functional interactions with atypical protein kinase C (aPKC) and its binding partner Partitioning defective protein 6 (Par-6) (*Kempkens et al., 2006*; *Morais-de-Sá et al., 2010*; *Tepass, 2012*; *Whitney et al., 2016*). These latter apical proteins act downstream of the small GTPase Cdc42 to promote phospho-dependent inhibition of lateral polarity proteins such as Lethal (2) giant larvae [L(2)gl] and Yrt (*Fletcher et al., 2012*; *Gamblin et al., 2014*; *Hutterer et al., 2004*; *Laprise and Tepass, 2011*; *Peterson et al., 2004*; *Tepass, 2012*). In return, L(2)gl and Yrt repress aPKC function to prevent the spread of apical characteristics to the lateral domain (*Bilder et al., 2003*; *Drummond and Prehoda, 2016*; *Fletcher et al., 2012*; *Gamblin et al., 2014*; *Hutterer et al., 2004*; *Laprise et al., 2006*; *Laprise et al., 2009*; *Tanentzapf and Tepass, 2003*; *Yamanaka et al., 2006*). Mutual antagonism between apical and lateral protein modules thus establishes a sharp boundary between their respective membrane domains (*Fletcher et al., 2012*; *Laprise and Tepass, 2011*). Cdc42 also activates p21-activated kinase 1 (Pak1), which has substrates in common with aPKC in fly epithelial cells (*Aguilar-Aragon et al., 2018*). However, it remains to be determined whether Pak1 targets the full complement of aPKC substrates, and whether phosphorylation by these kinases always has the same functional impact on target proteins and epithelial cell polarity. In particular, the functional relationship linking Pak1 and Yrt remains unexplored.

The mammalian Yrt orthologs EPB41L4B and EPB41L5 (also known as EHM2/LULU2 and YMO1/LULU, respectively) support cell migration and invasion, and EPB41L5 is also essential for the epithelial–mesenchymal transition (EMT) (*Handa et al., 2018*; *Hashimoto et al., 2016*; *Hirano et al., 2008*; *Jeong et al., 2019*; *Lee et al., 2007*; *Otsuka et al., 2016*; *Saller et al., 2019*; *Wang et al., 2006*; *Yu et al., 2010*). Given that Crb and human CRUMBS3 (CRB3) both repress EMT (*Campbell et al., 2011*; *Whiteman et al., 2008*), these observations are consistent with the aforementioned concept that fly and mammalian Yrt proteins repress the Crb/CRB3-containing apical machinery (*Laprise et al., 2006*; *Laprise and Tepass, 2011*). However, recent findings have shown that depletion of Crb or Yrt causes similar phenotypes in pupal wing cells. Specifically, DE-cadherin (DE-cad) staining is fragmented, and F-Actin and Myosin distribution is altered in the absence of these proteins (*Salis et al., 2017*). This suggests that Yrt controls cell contractility – a premise that awaits formal demonstration. Consistent with this model, it was shown that EPB41L4B and EPB41L5 promote apical constriction (*Nakajima and Tanoue, 2010*; *Nakajima and Tanoue, 2011*; *Nakajima and Tanoue, 2012*). Similar to EPB41L4B and EPB41L5, *Drosophila* Crb organizes the actomyosin cytoskeleton at cell–cell contacts, thereby supporting ZA integrity and tissue morphogenesis (*Letizia et al., 2011*; *Silver et al., 2019*; *Tepass, 1996*). Crb fulfils this function in ectodermal cells in part through the apical recruitment of the Guanine exchange factor (GEF) Cysts (Cyst), which activates the Rho1–Rho kinase (Rok)–Myosin pathway (*Arnold et al., 2017*; *Rubin et al., 2000*; *Silver et al., 2019*). In contrast, other studies have reported that Crb stabilizes adherens junctions by repressing Myosin activity in amnioserosa cells, a function that requires the FERM-domain protein Moesin (Moe) (*Flores-Benitez and Knust, 2015*). Moreover, Crb was shown to reduce the membrane residence time of Rok, thereby preventing the formation of a supracellular actomyosin cable in cells expressing high Crb levels at the salivary gland placode boundary where Crb distribution is anisotropic (*Röper, 2012*; *Sidor et al., 2020*). These conflicting observations show that the functional relationship linking Crb and Myosin is complex, and likely context-dependent. Whether, and how, Yrt modulates the multifaceted function of Crb in regulating cytoskeletal dynamics remains as an outstanding puzzle. Here, we show that Crb recruits Yrt to the apical membrane to induce Myosin-dependent apical tension in *Drosophila* epithelial cells. This association and the ensuing

increased Myosin activity is repressed by aPKC-dependent phosphorylation of Yrt. In contrast to aPKC, Pak1 maintains the pool of unphosphorylated Yrt via Protein phosphatase 2A (PP2A), thereby supporting Yrt-dependent cortical tension.

## Results

### Yrt increases Myosin-dependent cortical tension

It was previously shown that loss of Yrt expression increases cell diameter in pupal tissues (40), thus suggesting that Yrt may promote actomyosin-based contraction to define cell width. To test this hypothesis directly, we measured the retraction velocity after laser ablation of cell borders in the *Drosophila* embryonic epidermis, which reflects Myosin-dependent cortical tension (*Hutson et al., 2003*). Knockdown of *yrt* expression decreased the retraction velocity after laser ablation by 49% with respect to controls (*Figure 1A, B*; *Video 1*). In contrast, Yrt overexpression increased the retraction velocity after ablation by 43% (*Figure 1C, D*; *Video 2*). We quantified changes to the material properties of the apical cell surface using a Kelvin-Voigt model to measure the relaxation times of laser ablation experiments, proportional to the viscosity-to-elasticity ratio (*Fernandez-Gonzalez et al., 2009*). *yrt* knockdown and Yrt overexpression did not affect tissue viscoelasticity: relaxation times were 12 ± 1 seconds for *yrt* knockdown vs. 11 ± 2 seconds for controls, and 8 ± 1 seconds for Yrt overexpression vs. 10 ± 1 seconds for controls. Together, our data indicate that Yrt positively regulates cortical tension in the *Drosophila* embryonic epidermis.

Knockdown of *yrt* decreased cortical Spaghetti squash (Sqh; the *Drosophila* Myosin regulatory light chain) localization to cell boundaries without affecting Sqh intensity at the apex of epidermal cells from stage 14 of embryogenesis, a time point coinciding with the apical recruitment of Yrt (*Laprise et al., 2006*), whereas Yrt depletion had no major impact on Myosin distribution at earlier stages of development (*Figure 2A–D*). In addition, FLAG-Yrt expression resulted in apical enrichment of Sqh in adult ovarian follicle cells (*Figure 2E,F*). These data suggest that Yrt controls contractility at the apical domain. Consistent with this premise, mosaic overexpression of FLAG-Yrt induced apical constriction in the embryonic epidermis (*Figure 3A,B*) and the adult ovarian follicular epithelium (mRFP-positive cells; *Figure 3C,D*). Knockdown of *sqh* suppressed Yrt-induced apical constriction in (*Figure 3E,F*). Taken together, these results establish that Yrt promotes Myosin-based changes in cell mechanics to support cortical tension and/or apical constriction in various epithelial tissues, thereby providing putative molecular insights into how Yrt controls tissue morphogenesis (*Franke et al., 2005*; *Hoover and Bryant, 2002*; *Laprise et al., 2006*).

### aPKC counteracts Yrt-induced apical constriction by preventing the Crb-dependent apical recruitment of Yrt

aPKC phosphorylates Yrt on evolutionarily conserved residues (*Gamblin et al., 2014*). This prevents Yrt oligomerization, thereby antagonizing its ability to support lateral membrane stability and to restrict apical membrane growth (*Gamblin et al., 2018*). Expression of membrane-targeted aPKC (aPKC$^{CAAX}$) suppressed FLAG-Yrt-induced apical constriction (*Figure 4A–C,H*), showing that aPKC has a broad impact on Yrt function and also alters its ability to promote Myosin activity. However, aPKC$^{CAAX}$ had no effect on apical constriction induced by FLAG-Yrt$^{5A}$ in which the amino acids targeted by aPKC were replaced by non-phosphorylatable alanine residues [(*Gamblin et al., 2014*; *Figure 4D,E,H*)]. In addition, mutation of the aPKC phosphorylation sites into aspartic acids to generate the phosphomimetic Yrt mutant protein FLAG-Yrt$^{5D}$ (*Gamblin et al., 2014*) abolished the ability of Yrt to induce apical constriction (*Figure 4F,I*). Similarly, another FLAG-Yrt mutant protein unable to oligomerize [FLAG-Yrt$^{FWA}$; (*Gamblin et al., 2018*)] failed to sustain apical constriction (*Figure 4G,I*). These results argue strongly that aPKC targets Yrt directly to inhibit its ability to support apical constriction.

To gain further insights on how aPKC impacts on Yrt function in the follicular epithelium, we analyzed Yrt subcellular localization upon modulation of aPKC activity. We observed apical accumulation of Yrt in wild-type cells treated with a chemical inhibitor of aPKC or in *aPKC* knockdown cells (*Figure 5A,B*). These results show that aPKC normally acts to restrict Yrt apical localization, which depends on Crb in the embryonic epidermis and in pupal photoreceptor cells (*Laprise et al., 2006*). This is also the case in adult follicular epithelial cells, as Yrt failed to accumulate apically in *crb*

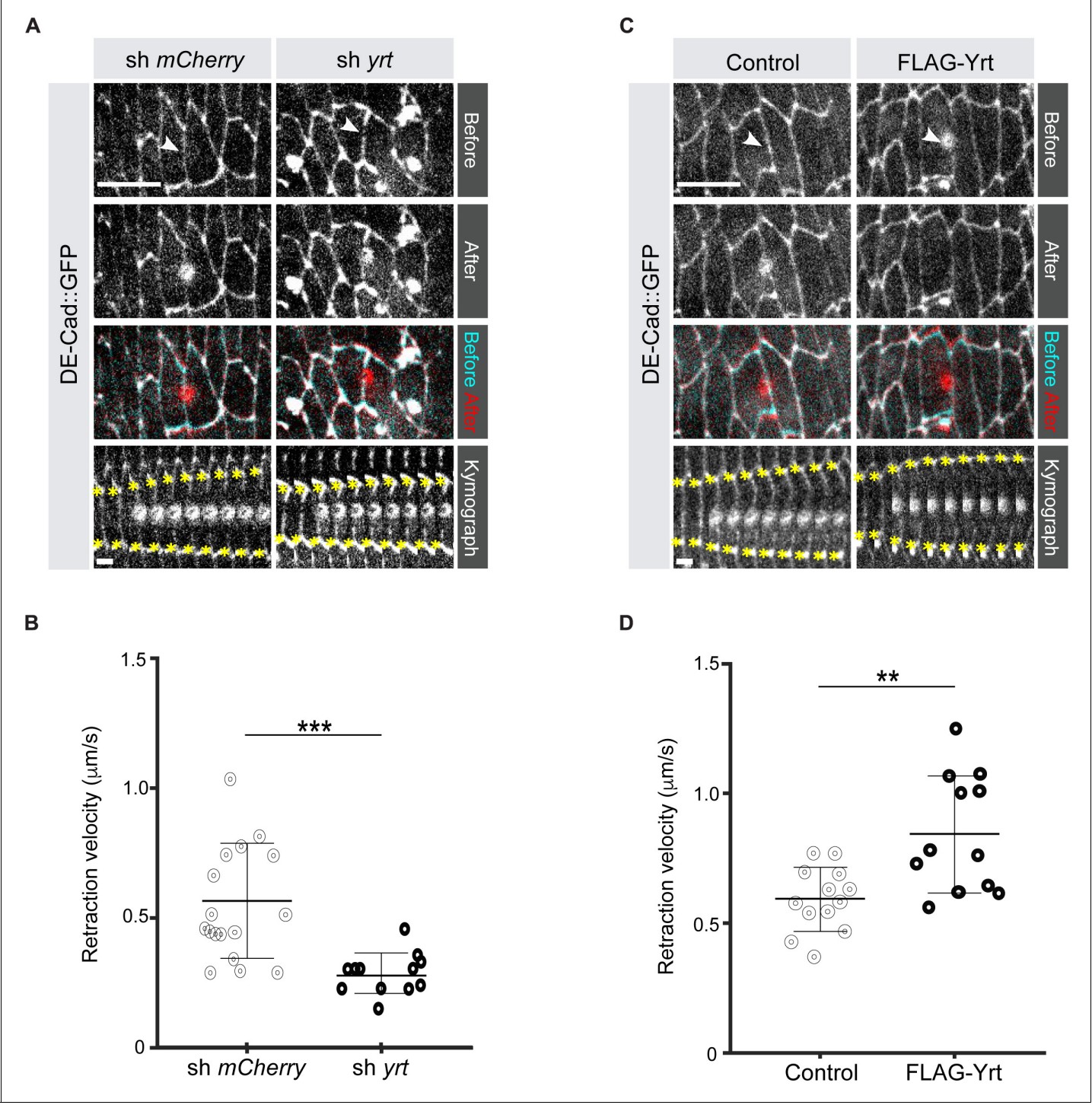

**Figure 1.** Yrt increases cortical tension. (**A**) Epidermal cells in stage 14 *Drosophila* embryos expressing DE-cad::GFP and shRNA against *mCherry* or *yrt* before (cyan in merge) and after (red in merge) laser ablation of a cell-cell interface (white arrowheads). Scale bar, 10 µm. Circular marks visible after ablation indicate where the UV laser passed through the vitelline membrane, creating autofluorescent holes. Bottom panel shows kymographs of interface retraction over time (scale bar, 4 s). Yellow asterisks indicate the position of the tracked vertices. (**B**) Retraction velocity after laser ablation in embryos expressing shRNA against *mCherry* (n = 17 junctions in 17 different embryos) or *yrt* (n = 12). (**C**) Epidermal cells in stage 14 *Drosophila* embryos expressing DE-cad::GFP and wild-type levels of Yrt expression (left panel; expression of LacZ as control) or overexpression of FLAG-Yrt (right panel) before (cyan in merge) and after (red in merge) ablation of a cell-cell interface (white arrowheads). Scale bar, 10 µm. (**D**) Retraction velocity after laser ablation in embryos with wild-type Yrt expression (n = 13 junctions in 13 different embryos) or overexpressing FLAG-Yrt (n = 12). In B and D, error bars indicate the standard deviation (s.d.), and the black bold line denotes the mean. ** p ≤ 0.01, *** p ≤ 0.001 (Mann-Whitney test).

*Figure 1 continued on next page*

*Figure 1 continued*

The online version of this article includes the following source data for figure 1:

**Source data 1.** Yrt increases retraction velocity.

mutant cells treated with the aPKC inhibitor (*Figure 5C,D*; *crb* mutant cells are positively labeled with GFP). Similarly, Yrt was absent from the apical membrane upon inhibition of aPKC in cells exclusively expressing a mutant Crb protein containing an amino acid substitution inactivating its FBM [Crb$^{Y10A}$; (*Figure 5E,F*; *Klebes and Knust, 2000*)], which prevents the direct association of Yrt and Crb proteins (*Laprise et al., 2006*). This suggests that Crb plays a pivotal role in Yrt-dependent apical constriction. Accordingly, FLAG-Yrt$^{5A}$ triggered apical constriction in wild-type cells, whereas it was unable to induce this phenotype in Crb-depleted cells or cells expressing specifically Crb$^{Y10A}$ (*Figure 5G–J*). Although our data establish that Crb and Yrt cooperate in promoting apical constriction, apical localization of Yrt in cells with suboptimal aPKC activity is not sufficient to induce prominently this phenotype (*Figure 5A,B*). Yrt and Moe, another FERM-domain protein binding to Crb, have opposite effects on Myosin-induced cortical tension [this study; (*Flores-Benitez and Knust, 2015*; *Laprise et al., 2006*; *Médina et al., 2002*; *Salis et al., 2017*)]. It is thus possible that the amount of Yrt relocalized to the apical domain upon inhibition of aPKC is not sufficient to outcompete Moe, whereas Yrt overexpression reaches the threshold required to displace most Moe proteins from Crb. Accordingly, inhibition of aPKC in *Moe* knockdown cells caused apical constriction, in contrast to what was observed in control cells (*Figure 6A,B*). In addition, expression of active Moe (*Karagiosis and Ready, 2004*) suppressed Yrt-induced apical constriction (*Figure 6C,D*), thereby confirming that these proteins are involved in a competitive functional interaction. Together, these results establish that aPKC precludes cortical tension by repressing the Yrt–Crb association, and that Moe antagonizes Yrt function.

## Pak1 and PP2A decrease Yrt phosphorylation and are required for Yrt-induced apical constriction

It was recently shown that aPKC and Pak1 share common phosphorylation targets, namely the polarity proteins L(2)gl, Bazooka (Baz), Par6, and Crb (*Aguilar-Aragon et al., 2018*). It was also proposed that aPKC and Pak1 act redundantly on these substrates (*Aguilar-Aragon et al., 2018*). However, it remains to be determined whether Pak1 impacts the function of other aPKC substrates such as Yrt. In addition, it is unclear if aPKC and Pak1 function is fully redundant in epithelial cells, or whether these kinases also have specific roles. Overexpression of Par6 together with aPKC$^{CAAX}$, which accumulated ectopically at the lateral membrane (*Figure 7A*), strongly reduced membrane localization of Yrt (*Figure 7A,B*). Expression of membrane-targeted Pak1 [Pak1$^{Myr}$; (*Noren et al., 2000*)] suppressed Yrt cortical release induced by aPKC and Par6 (*Figure 7A,B*). This result raises the intriguing possibility that Pak1 opposes aPKC function, a premise that we explored by investigating Yrt phosphorylation upon overexpression of these kinases. Overexpression of Par6 and aPKC$^{CAAX}$ is associated with an upward shift in the gel migration profile of Yrt (*Figure 7C*). This resulted from increased Yrt phosphorylation, as treatment of samples with the λ Phosphatase prior to electrophoresis abolished the impact of aPKC$^{CAAX}$ on Yrt gel mobility (*Figure 7C*; *Gamblin et al., 2014*). In contrast to aPKC, Pak1$^{Myr}$ decreased Yrt phosphorylation levels (*Figure 7C*). Given that Pak1 is a kinase, this observation is counterintuitive. However, it has been shown that Pak1 can activate the phosphatase PP2A (*Staser et al., 2013*), thus raising the possibility that the latter targets Yrt downstream of Pak1. Accordingly, Yrt phosphorylation levels were increased strongly in *PP2A-A* mutant embryos, or in wild-type embryos treated with the PP2A inhibitor Cantharidin (*Figure 7D,E*). Strikingly, inhibition of PP2A suppressed the impact of Pak1$^{Myr}$ expression on Yrt

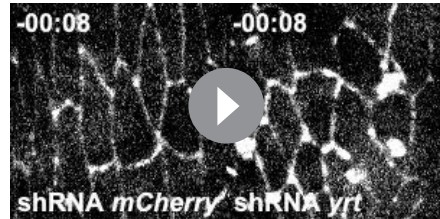

**Video 1.** *yrt* depletion decreases retraction velocity. Video of representative laser ablation experiments in control and *yrt* knockdown embryos.
https://elifesciences.org/articles/67999#video1

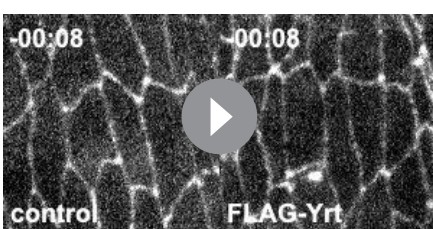

**Video 2.** Yrt overexpression increases retraction velocity. Video of representative laser ablation experiments in control and Yrt-overexpressing embryos.
https://elifesciences.org/articles/67999#video2

phosphorylation (*Figure 7F*), arguing that PP2A acts downstream of Pak1.

We then tested the functional impact of Pak1- and PP2A-dependent modulation of Yrt phosphorylation. Chemical inhibition of Pak1 or PP2A, or knockdown of their expression suppressed apical constriction induced by FLAG-Yrt (*Figure 8A–H*). Of note, FLAG-Yrt[5A] maintained its ability to induce apical constriction in the presence of Cantharidin, or when Pak1 or PP2A were depleted (*Figures 8C–D,E–H*). Pak1 and PP2A are thus dispensable for Yrt-induced apical constriction when aPKC target sites on Yrt are mutated to non-phosphorylatable residues. Altogether, these data suggest that Pak1 acts upstream of PP2A that antagonizes aPKC function by dephosphorylating Yrt, thereby allowing the latter to induce apical tension (*Figure 9*).

## Discussion

Myosin-mediated cell contractility impacts a broad range of cellular processes driving tissue morphogenesis, including apical constriction, convergent extension, and ZA establishment and maintenance (*Miao and Blankenship, 2020*; *Perez-Vale and Peifer, 2020*). Our data establish that Yrt increases Myosin-dependent cortical tension in embryonic and adult tissues, thereby providing further insights into how Yrt contributes to organize epithelial tissue architecture. Similarly, a recent study showed that loss of Yrt expression increases cell diameter in pupal wing cells (*Salis et al., 2017*), thus suggesting that Yrt controls the actomyosin network throughout the fly life cycle. This function is evolutionarily conserved, as the mammalian Yrt proteins EPB41L4B and EPB41L5 impact actomyosin organization and stimulate apical constriction (*Nakajima and Tanoue, 2010*; *Nakajima and Tanoue, 2011*; *Nakajima and Tanoue, 2012*). Regulation of actomyosin contractility by Yrt likely impacts ZA stability, as suggested by the fragmented distribution of ZA components in Yrt-depleted cells (*Salis et al., 2017*). This is in line with the fact that Yrt-induced apical constriction depends on Crb, which is also required for organization of junctional Myosin assembly and ZA stability (*Pocha and Knust, 2013*; *Silver et al., 2019*; *Tepass, 1996*).

Our findings showing that Crb and Yrt cooperate in promoting cortical tension may seem at odds with previous studies demonstrating that Yrt inhibits the ability of Crb to support apical membrane growth (*Laprise et al., 2006*; *Laprise et al., 2009*). We propose that Yrt is not an inhibitor of Crb function, but rather acts downstream of Crb to specifically promote actomyosin contractility at the expense of other Crb-dependent roles such as apical membrane growth. It is possible that the recruitment of Yrt causes steric hindrance on the short cytoplasmic tail of Crb, thereby preventing the binding of Crb effectors mediating apical membrane growth. In addition, we found that Yrt competes with Moe, which binds to Crb and decreases cortical tension (*Flores-Benitez and Knust, 2015*; *Médina et al., 2002*; *Salis et al., 2017*). Our model is that the numerous functions of Crb are specified by the presence or absence of different FERM domain proteins, which competitively binds to Crb. This paradigm is further supported by previous findings showing that Crb controls cell growth through recruitment of the FERM domain protein Expanded (Ex) (*Ling et al., 2010*; *Robinson et al., 2010*). Thus, fine regulation of the incorporation of different FERM-domain proteins within distinct Crb complexes most likely dictates the biological output downstream of Crb. This ordered equilibrium may explain the functional plasticity of Crb, which can both increase or decrease Myosin-dependent contractility to accommodate dynamic regulation of epithelial cell polarity, cell–cell adhesion, and tissue morphogenesis (*Bajur et al., 2019*; *Flores-Benitez and Knust, 2015*; *Letizia et al., 2011*; *Röper, 2012*; *Sidor et al., 2020*; *Silver et al., 2019*).

Our discoveries indicate that aPKC prevents the Crb–Yrt association through direct phosphorylation of Yrt. Thereby, aPKC excludes Yrt from the apical membrane and represses Yrt-induced apical constriction. aPKC also inhibits the ability of Yrt to restrict Crb-dependent apical membrane growth (*Gamblin et al., 2014*; *Gamblin et al., 2018*). This is consistent with the aforementioned model, and

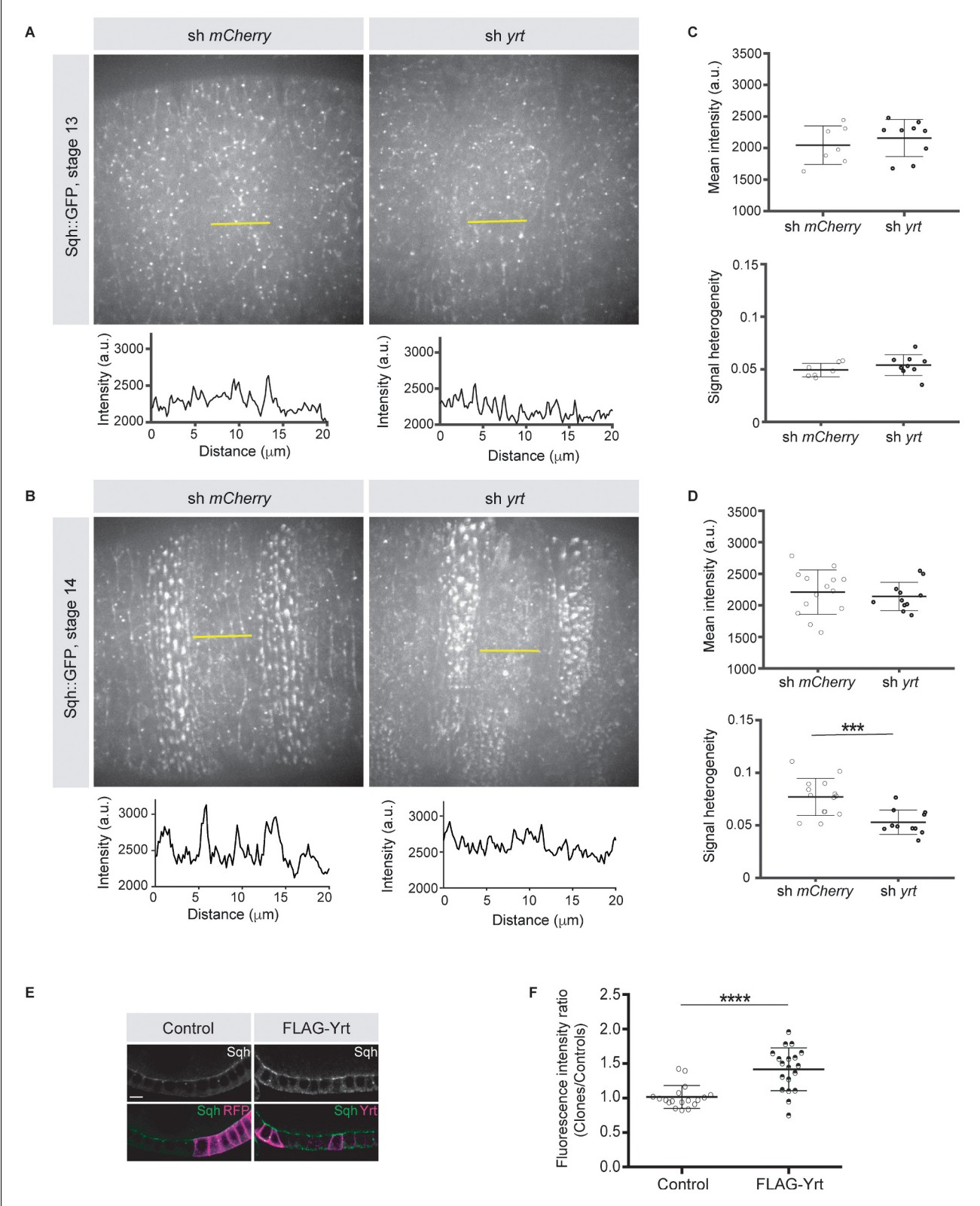

**Figure 2.** Yrt promotes apical enrichment of Myosin. (**A, B**) Expression of Sqh:GFP and shRNA targeting either *mCherry* or *yrt* in the embryonic epidermis at stages 13 (**A**) and 14 (**B**). Intensity vs. distance is plotted for indicated yellow lines. (**C, D**) Mean Sqh intensity and heterogeneity of Sqh signal for antero-posterior linescans in stage 13 embryos (**C**; *n* = seven sh *mCherry*, nine sh *yrt*) and in stage 14 embryos (**D**; *n* = 14 sh *mCherry*, 11 sh *yrt*). Error bars indicate the standard deviation (s.d.), and black lines denote the mean. *** p < 0.001 (Mann-Whitney test). (**E**) Mosaic expression of

*Figure 2 continued on next page*

Figure 2 continued

FLAG-Yrt in follicular epithelial cells constitutively expression Sqh::GFP, which was visualized by immunofluorescence. FLAG-Yrt-expressing cells (right panels) are labeled with mRFP, which was co-expressed with LacZ in control clones (left panels). Scale bar represents 5 μm. (F) Quantification of the fluorescence intensity of apical Sqh::GFP in cells expressing the exogenous proteins described in E (stage 5 and 6 follicles were used for quantification). Results were calculated as a ratio between FLAG-Yrt-expressing cells and control cells in the same follicle ($n$ = 18 follicles for the control and $n$ = 19 follicles for FLAG-Yrt). **** $p \leq 0.0001$ (one-way ANOVA), error bars indicate s.d.; bold black lines denote the mean in **F**.

The online version of this article includes the following source data for figure 2:

**Source data 1.** Yrt controls Myosin distribution from stage 14 of embryogenesis.
**Source data 2.** Yrt increases apical Sqh in the follicular epithelium.

---

with previous reports documenting functional cooperation between aPKC and Crb in establishing the apical domain (*Fletcher et al., 2012*; *Sotillos et al., 2004*; *Tepass, 2012*; *Walther and Pichaud, 2010*). We have shown that phosphorylation of Yrt on aPKC target sites prevents Yrt self-association, and that oligomerization-defective Yrt mutant proteins are unable to bind Crb (*Gamblin et al., 2018*). Together, these studies establish that the phospho-dependent repression of Yrt oligomerization is a key mechanism by which aPKC controls apical–basal polarity, cortical tension, and epithelial cell architecture. Our data also show that ectopic localization of aPKC to the lateral domain dislodges Yrt from the membrane, indicating that aPKC controls both Crb-dependent and Crb-independent membrane localization of Yrt. One possibility is that phosphorylation of Yrt by aPKC generates electrostatic repulsion with negatively charged membrane phospholipids, as described for other aPKC substrates (*Bailey and Prehoda, 2015*).

Cdc42 is a key regulator of epithelial cell polarity acting upstream of aPKC and Pak1 (*Aguilar-Aragon et al., 2018*; *Pichaud et al., 2019*). These kinases share common substrates and have redundant functions in apical–basal polarity regulation (*Aguilar-Aragon et al., 2018*). However, Yrt sequence does not contain a Pak1 consensus motif. Moreover, we found that expression of membrane-targeted Pak1 reduces Yrt phosphorylation levels, whereas membrane-bound aPKC has the opposite effect. This argues that Yrt is not a direct substrate of Pak1, and that this kinase does not phosphorylate the full complement of aPKC substrates within the polarity protein network. Finally, while Pak1 is required for Yrt-induced apical constriction, aPKC inhibits this function of Yrt. In the light of all of this evidence, we propose that Pak1 promotes the dephosphorylation of aPKC target sites on Yrt to support its functions. We provide evidence that Pak1 acts through the phosphatase PP2A to achieve this function in polarized epithelial cells. PP2A thus has a broad role in cell polarity, as it opposes aPKC and Par-1 signaling to control polarization of neuroblasts and photoreceptor cells (*Krahn et al., 2009*; *Nam et al., 2007*; *Ogawa et al., 2009*). In these cell types, PP2A targets Par6 and/or Baz. This implies that Pak1 may also decrease the phosphorylation level of Baz and/or Par6 in epithelial cells, although the kinase domain of Pak1 can phosphorylate Baz and Par6 in vitro (*Aguilar-Aragon et al., 2018*). At this point, a cell type-specific or context-dependent regulation of Baz and Par-6 by Pak1 cannot be ruled out. Alternatively, it is possible that Pak1 can both increase or decrease the phosphorylation of specific polarity proteins in regulatory feedback loops, thereby ensuring the delicate equilibrium required for epithelial tissue homeostasis. Combining our data with previous findings (*Aguilar-Aragon et al., 2018*) suggests that activation of aPKC and Pak1 downstream of Cdc42 has redundant activity on a subset of polarity proteins [e.g. Par6, L(2)gl, Baz], while having opposite effects on Yrt and apical tension. Identification of the molecular mechanisms selecting the engagement of aPKC or Pak1 downstream of Cdc42 remains an outstanding puzzle, solving of which will provide crucial insights into the understanding of epithelial cell polarity, cortical tension, and tissue morphogenesis.

Overall, our work reveals a novel mechanism regulating cortical tension at the apical domain, which plays a crucial role in developing and adult epithelia. We also refine our understanding of the functional relationship linking Crb and Yrt by establishing that the latter contributes to the plasticity of Crb function rather than acting as an obligatory inhibitor of the apical protein machinery. We also highlight that the roles of aPKC and Pak1 are not fully redundant in epithelial cells, and that these kinases have opposite effects on cortical tension. We also provide evidence that Pak1 can decrease protein phosphorylation in epithelial cells through PP2A, thereby consolidating the broad role played by this phosphatase in cell polarity.

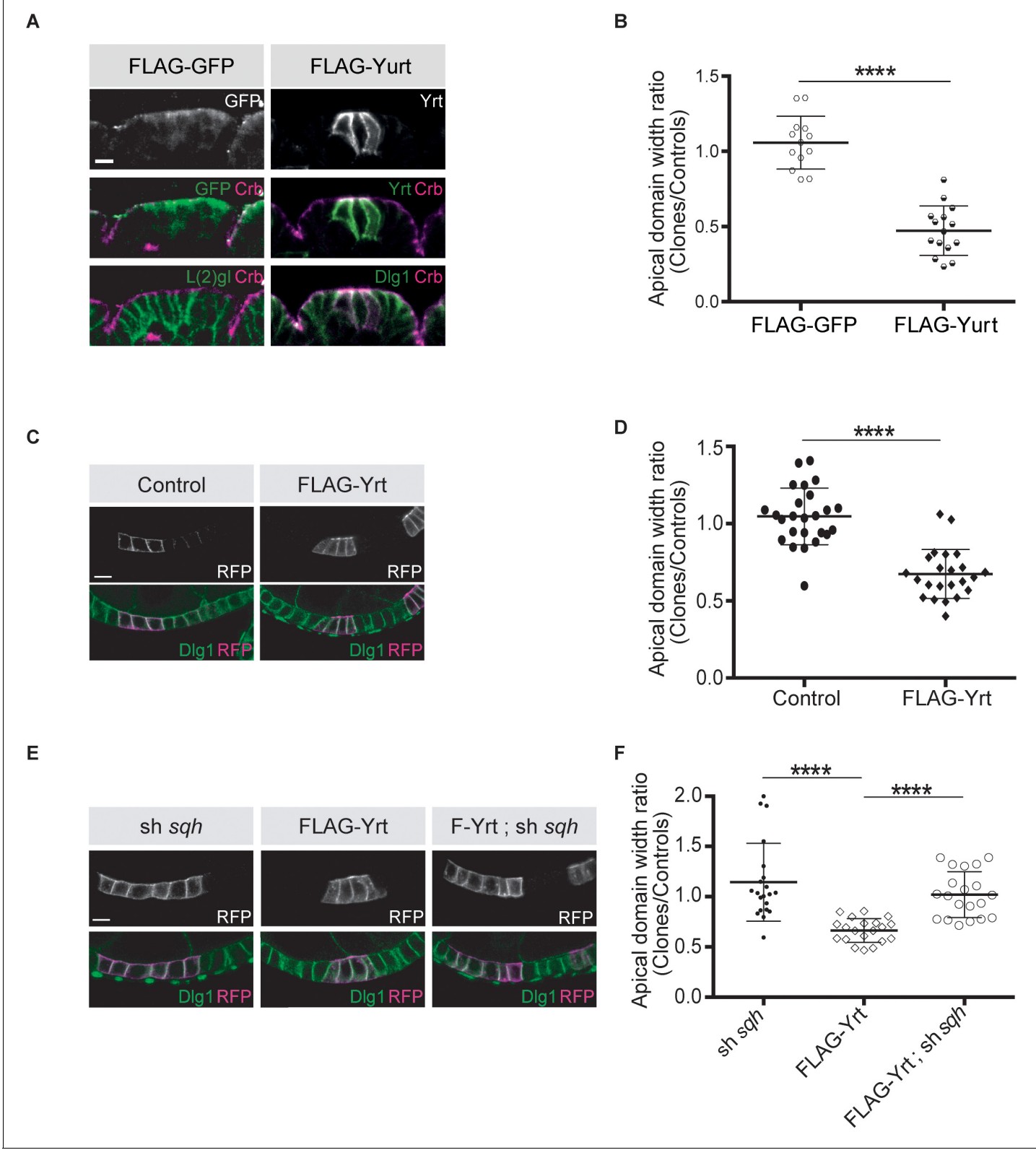

**Figure 3.** Yrt promotes apical constriction. (**A**) Mosaic expression of FLAG-GFP or FLAG-Yrt in the embryonic epidermis at stage 14. Crb immunostaining (magenta) marks the apical domain, whereas immunolabeling of L(2)gl or Discs large 1 (Dlg1) label the lateral membrane. (**B**) Quantification of apical domain width (*n* = 13 for FLAG-GFP and *n* = 15 for FLAG-Yrt; images were acquired from seven individual embryos for each genotype). Results are expressed as the ratio between the width of cells expressing the transgenes and the width of control cells in the same segment

*Figure 3 continued on next page*

*Figure 3 continued*

(**** p ≤ 0.0001). Scale bar represents 5 µm. (**C**) Clonal expression of membrane-targeted RFP (mRFP) and LacZ (control; left panels), or mRFP and FLAG-Yrt (F-Yrt; right panels) in the follicular epithelium. Immunostaining of mRFP highlights cells expressing the transgenes, whereas immunolabeling of endogenous Dlg1 (green) marks the lateral membrane of all epithelial cells. (**D**) Quantification of apical domain width of cells expressing the transgenes listed in **C**. Results are expressed as the ratio between the width of cells expressing the transgenes and the width of control cells in the same follicle. (**E**) Mosaic expression of a shRNA targeting *sqh* (left panels), of FLAG-Yrt (center panels), or expression of both sh-*sqh* and FLAG-Yrt (right panels). mRFP was co-expressed with these constructs, and used to label positive cells. (**F**) Quantification of the apical diameter of cells expressing the transgenes indicated in **E**. Results are expressed as a ratio to the width of control cells. In **B**, **D** and **F**, error bars indicate s.d.; bold black lines denote the mean; *n* ≥ 20; **** p ≤ 0.0001 (Student's t-test was used in **B**, one-way ANOVA for **D**, **F**). Follicles shown in **C** and **E** were at stage 5. Scale bars represent 5 µm.

The online version of this article includes the following source data for figure 3:

**Source data 1.** Yrt promotes apical constriction.

# Materials and methods

## *Drosophila* genetics

The UAS-GAL4 system was employed to drive the expression of transgenes (*Brand et al., 1994*). Mosaic analysis in the follicular epithelium was performed using the Flippase recognition target (FRT)/Flippase (FLP) system in which the FLP recombinase was under the control of a heat shock

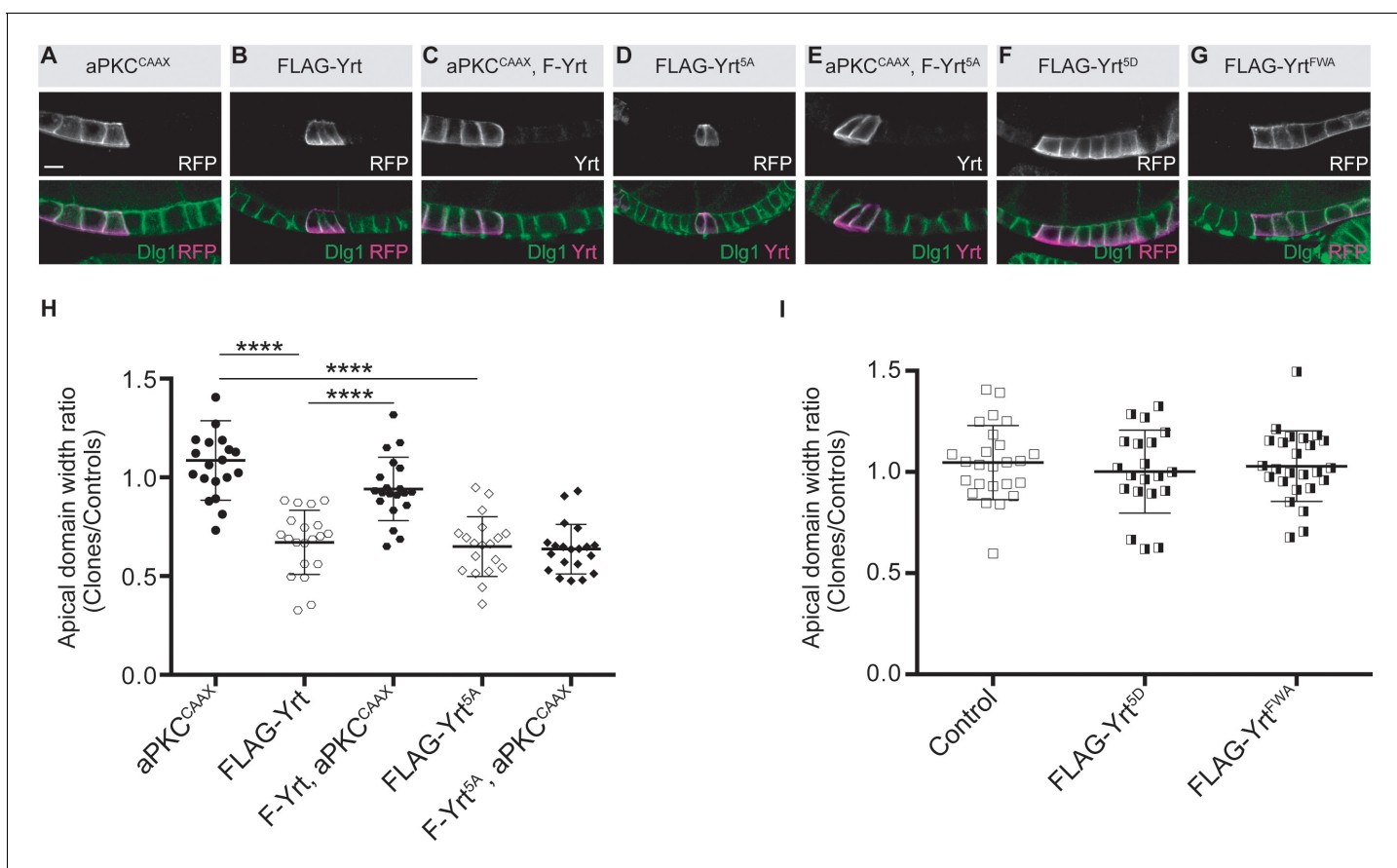

**Figure 4.** aPKC represses Yrt-induced apical constriction. (**A–G**) Clonal expression of the indicated proteins in the follicular epithelium. Clones were positively labeled with mRFP or Yrt staining. (**H**, **I**) Quantification of the apical diameter of cells expressing the transgenes indicated in **A–E** (**H**) or **F**, **G** (**I**). Results are expressed as a ratio to the width of control cells in the same follicle (stage 5 and 6 follicles were used for quantification). Scale bars represent 5 µm. **** p ≤ 0.0001 (one-way ANOVA), error bars indicate s.d.; bold black lines denote the mean, *n* ≥ 19.

The online version of this article includes the following source data for figure 4:

**Source data 1.** aPKC represses Yrt function.

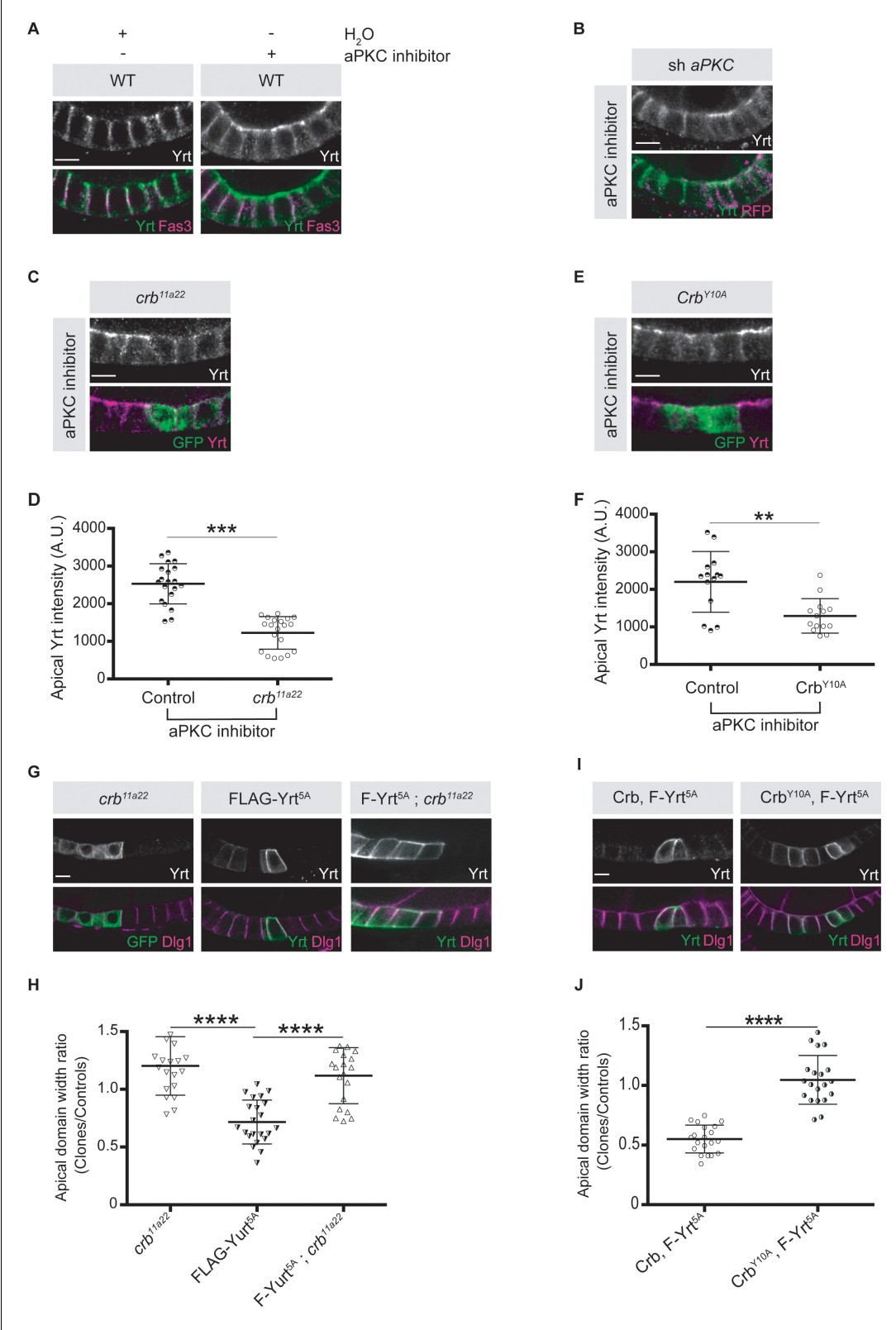

**Figure 5.** Crb is essential for Yrt-dependent apical constriction. (**A**) Panels depict immunofluorescence of Yrt (green in merge) and Fasciclin 3 (Fas3; lateral marker, magenta) in the follicular epithelium treated for 2 hr with the aPKC inhibitor CRT-006-68-54. (**B**) *aPKC* knockdown cells (mRFP positive) were immunostained for Yrt. (**C**) *crb*[11A22] (null allele) homozygous mutant clones were produced in adult *crb*/+ female flies (mutant clones are GFP positive). Dissected ovaries were incubated with the aPKC inhibitor CRT-006-68-54 prior to fixation and Yrt immunostaining. (**D**) Quantification of apical

*Figure 5 continued on next page*

*Figure 5 continued*

Yrt intensity in control or *crb* mutant cells within the same follicle in presence of the aPKC inhibitor CRT-006-68-54. (E) Analysis of Yrt localization in control or *crb*[11A22] mutant cells expressing exogenous Crb[Y10A] (GFP-positive cells) exposed to the aPKC inhibitor CRT-006-68-54. (F) Quantification of apical Yrt staining in control and *crb* null cells expressing exogenous Crb[Y10A] treated with the aPKC inhibitor. (G) FLAG-Yrt[5A] was specifically expressed in *crb* mutant cell clones (right panels). Mosaic expression of FLAG-Yrt[5A] in control cells or *crb* mutant clones were used as controls (middle and left panels, respectively). Clones were labeled with GFP (left panels) or by Yrt immunostaining (middle and right panels). Dlg1 staining was used to label the lateral membrane. (H) Quantification of the apical domain width of cells expressing the transgenes listed in G. (I) Immunostaining of Yrt and Dlg1 in follicular epithelial cells expressing FLAG-Yrt[5A] and exogenous wild-type Crb (left panels), or FLAG-Yrt[5A] together with Crb[Y10A] (right panels). (J) Quantification of the apical domain width of cells expressing the transgenes listed in I. Results are expressed as the ratio between the width of cells expressing the transgenes and the width of control cells in the same follicle. Stage three follicles were depicted in A and B, whereas panels C, E, G, and I display stage five follicles. In D, F, H, and J error bars indicate s.d.; bold black lines denote the mean; $n \geq 20$; ** $p \leq 0.01$, *** $p \leq 0.001$, **** $p \leq 0.0001$ (one-way ANOVA). All scale bars represent 5 µm.

The online version of this article includes the following source data for figure 5:

**Source data 1.** Crb cooperates with Yrt.

promoter (*hsFLP*) (*Chou and Perrimon, 1996*; *Golic and Lindquist, 1989*). Newly hatched females were heat-shocked for 3 hr at 37°C on two consecutive days. Ovaries were dissected and processed 48 hr after the second heat shock. *Tables 1* and *2* depict the stock list and the complete genotype of flies used in this study, respectively.

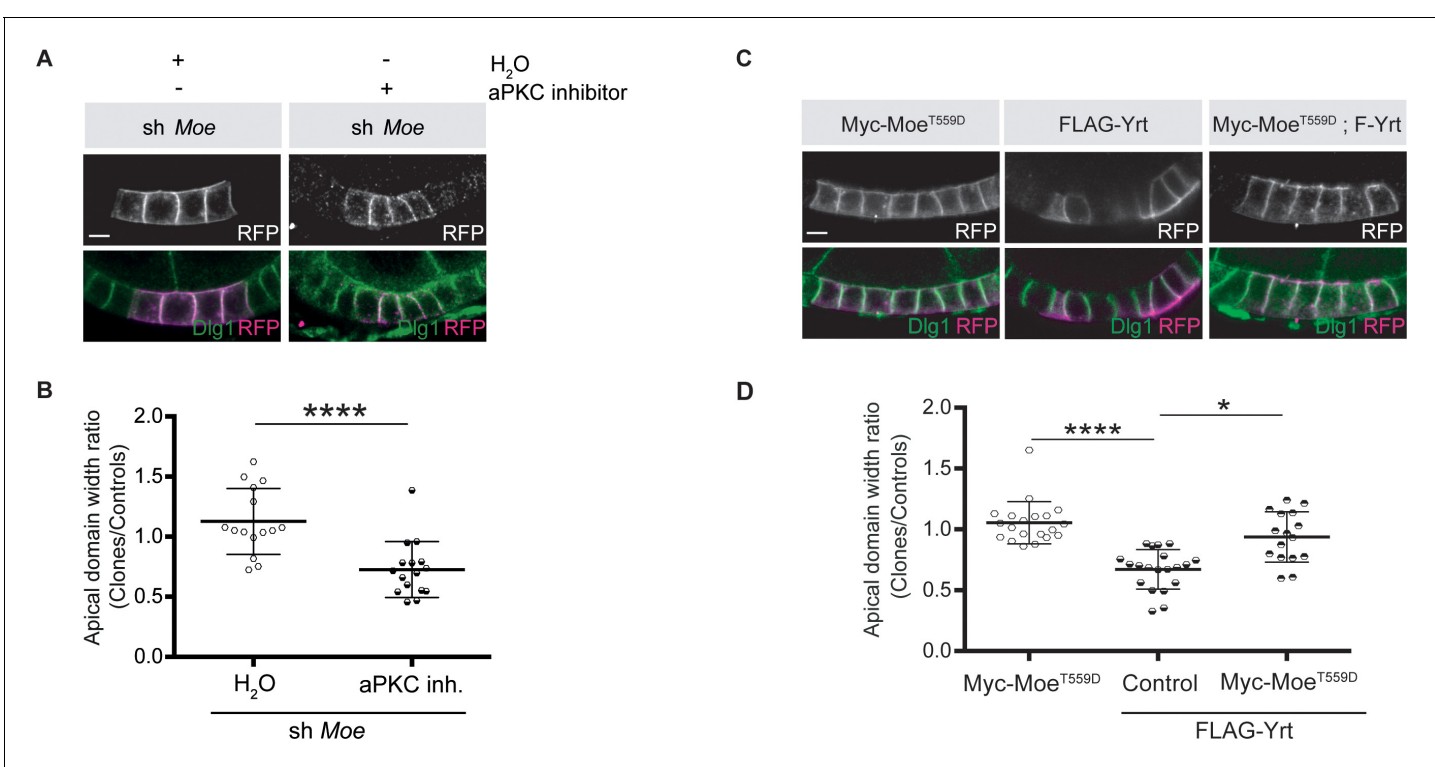

**Figure 6.** Moe suppresses Yrt-induced apical constriction. (A) Mosaic knockdown of *Moe* in absence or presence of the aPKC inhibitor CRT-006-68-54. Clones are positively labeled with membrane-targeted RFP. (B) Quantification of apical domain width of knockdown cells in absence or presence of the aPKC inhibitor CRT-006-68-54. Results were calculated as a ratio between clonal cells and control cells in the same follicle. (C) Clonal expression of FLAG-Yrt, Myc-Moe[T559D] (*Karagiosis and Ready, 2004*), or both proteins (clones are labeled with mRFP). (D) Quantification of the apical domain width of cells expressing the transgenes listed in C. Results are expressed as the ratio between the width of cells expressing transgenes and the width of control cells in the same follicle (quantification was performed using stage five follicles). In B and D, error bars indicate s.d.; bold black lines denote the mean; $n \geq 18$; * $p \leq 0.05$, **** $p \leq 0.0001$ (one-way ANOVA). All scale bars represent 5 µm.

The online version of this article includes the following source data for figure 6:

**Source data 1.** Moe and Yrt show antagonistic function.

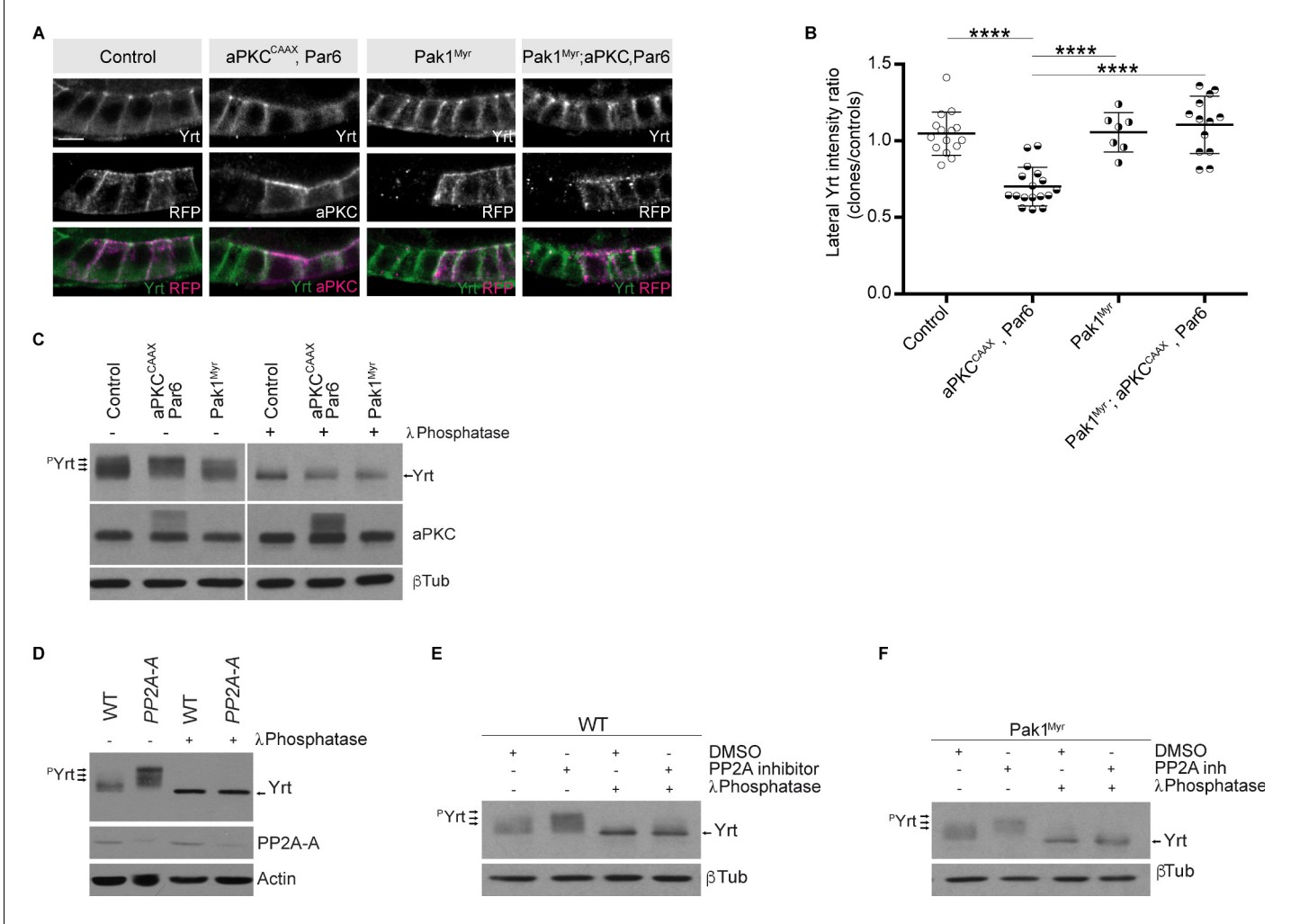

**Figure 7.** Pak1 and PP2A control the phosphorylation level of Yrt. (**A**) Yrt immunostaining in control follicular epithelial cells or clones of cells expressing aPKC[CAAX] together with Par6, Pak1[Myr], or aPKC[CAAX] with Par6 and Pak1[Myr]. Clones were positively labeled with RFP or aPKC immunostaining in stage four follicles. Scale bar represents 5 μm. (**B**) Quantification of lateral Yrt staining intensity in genotypes described in A. Error bars indicate s.d.; bold black lines denote the mean; $n \geq 16$; **** $p \leq 0.0001$ (one-way ANOVA). (**C**) Control embryos, embryos overexpressing Par-6 and aPKC[CAAX], or embryos expressing Pak1[Myr] were homogenized. Samples were incubated or not with the λ phosphatase and processed for SDS-PAGE. Western blotting using anti-Yrt antibodies showed the migration profile of phosphorylated ([P]Yrt) or unphosphorylated Yrt, whereas β-Tubulin (βTub) was used as loading control. (**D**) Western blots showing the migration profile of Yrt ([P]Yrt stands for phosphorylated Yrt) extracted from wild-type (WT) or *PP2A-A* mutant embryos. The expression level of PP2A-A is also shown, and Actin was used as loading control. (**E**) Wild-type embryos were treated or not with the PP2A inhibitor Cantharidin, homogenized, and processed for SDS-PAGE. Western blotting using control samples shows the phosphorylation levels of Yrt ([P]Yrt), whereas samples incubated with λ Phosphatase prior to gel electrophoresis show unphosphorylated Yrt. β-Tubulin (βTub) was used as loading control. (**F**) Embryos overexpressing Pak1[Myr] were treated or not with PP2A inhibitor and homogenized. Western blotting using anti-Yrt antibodies showed the migration profile of phosphorylated ([P]Yrt) or unphosphorylated Yrt. λ Phosphatase was used as a positive control of Yrt dephosphorylation, and β-Tubulin (βTub) was used as loading control.

The online version of this article includes the following source data for figure 7:

**Source data 1.** aPKC and Pak1 control Yrt localization.
**Source data 2.** aPKC and Pak1 control Yrt phosphorylation.
**Source data 3.** PP2A dephosphorylates Yrt.
**Source data 4.** Pak1 acts upstream of PP2A.
**Source data 5.** Loading controls.
**Source data 6.** Annotated blots.

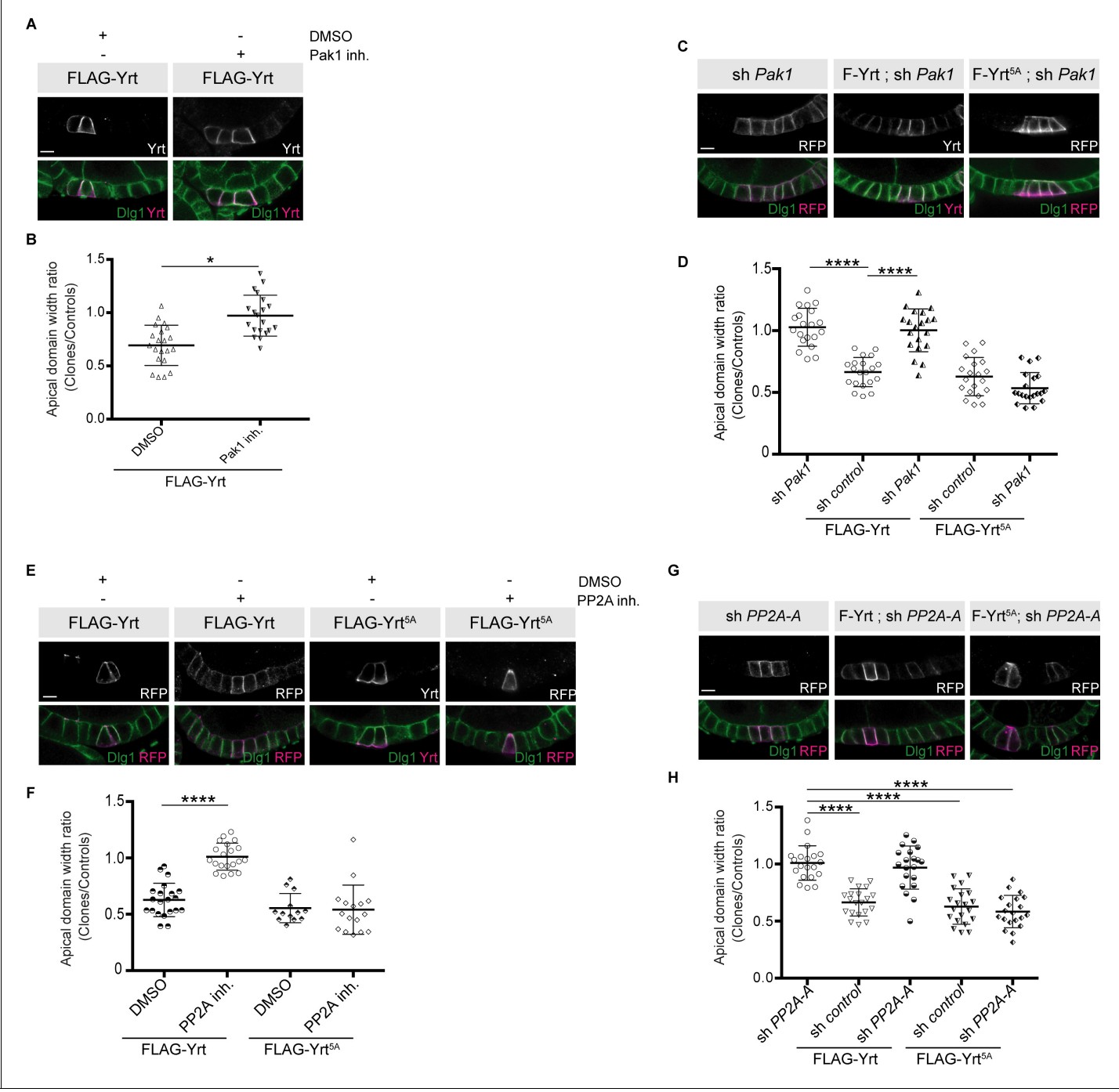

**Figure 8.** Pak1 and PP2A are essential for Yrt-induced apical constriction. (**A**) Stage four follicles clonally overexpressing FLAG-Yrt were incubated with the vehicle DMSO or the Pak1 inhibitor (IPA-3; Pak1 inh.) prior to fixation and immunostaining with antibodies directed against Yrt and Dlg1. (**B**) Quantification of the apical diameter of cells expressing transgenes indicated in A incubated or not with IPA-3. Results are expressed as the ratio between the width of cells expressing the transgenes and the width of control cells in the same follicle. (**C**) Immunostaining of Dlg1 in follicles displaying mosaic expression of a shRNA targeting *Pak1* and the indicated FLAG-Yrt constructs. Clones were labeled with mRFP or Yrt staining. (**D**) Quantification of the diameter of cells expressing the exogenous proteins described in **C**. Results were calculated as a ratio between transgene expressing cells and control cells within the same follicle. (**E**) Follicles containing cell clones expressing FLAG-Yrt or FLAG-Yrt[5A] (mRFP-positive cells) were incubated or not with the PP2A inhibitor Cantharidin (PP2A inh.) prior to immunostaining with indicated antibodies. (**F**) Quantification of the apical diameter of cells expressing transgenes indicated in **E**. Results are expressed as the ratio between the width of cells expressing the transgenes and the width of control cells in the same follicle. (**G**) Panels depict follicular epithelium displaying mosaic knockdown of *PP2A-A* (left panels), mosaic expression of FLAG-Yrt (middle panels), or the combination of the indicated FLAG-Yrt constructs with a shRNA targeting *PP2A-A* (middle and right panels).

*Figure 8 continued on next page*

*Figure 8 continued*

Immunostaining of mRFP or Yrt highlights cells expressing the transgenes, whereas immunolabeling of endogenous Dlg1 shows the lateral domain of all epithelial cells. (H) Quantification of the apical diameter of cells expressing transgenes indicated in G. Results are expressed as the ratio between the width of cells expressing the transgenes and the width of control cells in the same follicle. B, D, F, and H, error bars indicate s.d.; bold black lines denote the mean; $n \geq 16$; * $p \leq 0.05$, **** $p \leq 0.0001$ (one-way ANOVA). (C, F and G) imaged follicles were at stages 5 or 6. All scale bars represent 5 µm.

The online version of this article includes the following source data for figure 8:

**Source data 1.** Pak1 and PP2A promote apical constriction.

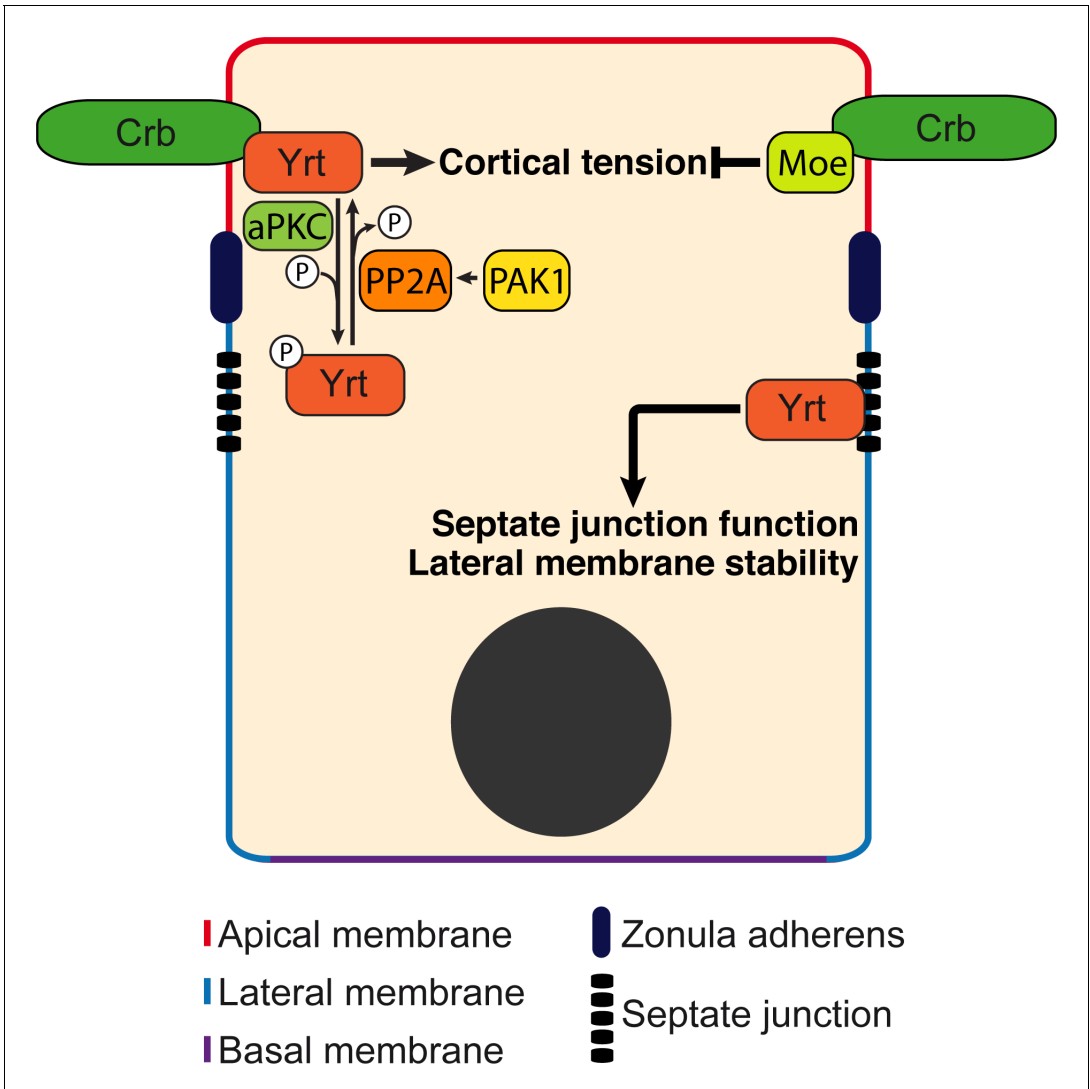

**Figure 9.** Model: Yrt is a multifunctional protein promoting cortical tension downstream of Crb. Yrt is localized to the lateral membrane where it prevents the spread of apical characteristics in differentiating epithelial cells and controls the occluding function of septate junctions in fully differentiated cells (*Laprise et al., 2006*; *Laprise et al., 2009*; *Laprise and Tepass, 2011*; *Sollier et al., 2015*). Yrt also occupies the apical domain owing to its direct interaction with Crb (*Laprise et al., 2006*). Yrt promotes Crb-dependent cortical tension (this study) in contrast to Moe that represses Myosin function downstream of Crb (*Flores-Benitez and Knust, 2015*; *Salis et al., 2017*). Hence, these proteins contribute to the functional plasticity of Crb, and a fine regulation of their association with Crb is thus required to define and stabilize the functional architecture of epithelial cells. Our data indicate that aPKC plays a key role in this process through phospho-dependent exclusion of Yrt from the apical domain. In contrast, Pak1 promotes Yrt dephosphorylation through activation of PP2A. The Pak1–PP2A module, which opposes aPKC function, is thus essential for Yrt-induced contractility. The equilibrium between aPKC and Pak1–PP2A activities thus balances Yrt and Crb functions.

**Table 1.** List of fly stocks used in this study.

| Genotype | Source | Identifier |
|---|---|---|
| *D. melanogaster: UAS-FLAG-Yurt* | Reference (*Gamblin et al., 2014*) | N/A |
| *D. melanogaster: UAS-FLAG-Yurt[5A]* | Reference (*Gamblin et al., 2014*) | N/A |
| *D. melanogaster: UAS-FLAG-Yurt[FWA]* | Reference (*Gamblin et al., 2018*) | N/A |
| *D. melanogaster: UAS-FLAG-Yurt[5D]* | Reference (*Gamblin et al., 2014*) | N/A |
| *D. melanogaster: UAS-sh yurt* | *Drosophila* Bloomington Stock Centre | # 36118 |
| *D. melanogaster: UAS-sh squash* | *Drosophila* Bloomington Stock Centre | # 32439 |
| *D. melanogaster: UAS-aPKC[CAAX]* | Reference (*Sotillos et al., 2004*) | N/A |
| *D. melanogaster: UAS-aPKC[CAAX], Par6* | Gift from T. Harris (*Harris and Tepass, 2010*) | N/A |
| *D. melanogaster: UAS-sh mCherry* | *Drosophila* Bloomington Stock Centre | # 35785 |
| *D. melanogaster: FRT82b, crb[11a22]* | *Drosophila* Bloomington Stock Centre | # 3448 |
| *D. melanogaster: UAS-sh aPKC* | *Drosophila* Bloomington Stock Centre | # 34332 |
| *D. melanogaster: UAS-sh Pak1* | *Drosophila* Bloomington Stock Centre | # 41714 |
| *D. melanogaster: UAS-Pak[myr]* | *Drosophila* Bloomington Stock Centre | # 8804 |
| *D. melanogaster: foscrb[Y10A]; FRT82b, crb[11a22]* | Reference (*Klose et al., 2013*) | N/A |
| *D. melanogaster: UAS-LacZ* | *Drosophila* Bloomington Stock Centre | # 8529 |
| *D. melanogaster: sh Pp2a-A* | *Drosophila* Bloomington Stock Centre | # 55050 |
| *D. melanogaster: endo-E-Cadherin::GFP* | Reference (*Huang et al., 2009*) | N/A |
| *D. melanogaster: sqh-sqh::GFP* | Reference (*Royou et al., 2004*) | N/A |
| *D. melanogaster: matαtub67;15 GAL4* | Reference (*Staller et al., 2013*) | N/A |
| *D. melanogaster: daughterless-GAL4* | *Drosophila* Bloomington Stock Centre | # 27608 |
| *D. melanogaster: hs-FLP;; tub-FRT-Gal80-FRT-Gal4, UAS–mRFP* | Gift from Y. Bellaiche, Institut Curie, PSL Research University, Paris, France. | N/A |
| *D. melanogaster: hs-FLP; tub-GAL4, UAS-GFP; FRT82b, tub-GAL80* | Gift from N. Perrimon, Harvard Medical School, Boston, USA | N/A |
| *D. melanogaster: hs-FLP; sqh-Sqh::GFP; tub>Gal80>Gal4, UAS–mRFP* | Gift from Y. Bellaiche, Institut Curie, PSL Research University, Paris, France. | N/A |
| *D. melanogaster: wild type* | *Drosophila* Bloomington Stock Centre | # 25210 |
| *D. melanogaster: PP2A-A[EP2332] / Cyo* | *Drosophila* Bloomington Stock Centre | # 1704 |
| *D. melanogaster: Patched-GAL4* | *Drosophila* Bloomington Stock Centre | # 65661 |
| *D. melanogaster: UAS-myc-Moesin[T559D]* | *Drosophila* Bloomington Stock Centre | # 8630 |
| *D. melanogaster: UAS-sh Moesin* | *Drosophila* Bloomington Stock Centre | # 8629 |
| *D. melanogaster: UAS-FLAG-GFP* | Reference (*Gamblin et al., 2018*) | N/A |

## Laser ablation for measurement of mechanical properties

Stage 14 embryos were dechorionated for 2 min in 50% bleach, washed with water, and glued on a coverslip using heptane glue (*Scepanovic et al., 2021*). Ablations were induced using a pulsed Micropoint N$_2$ laser (Andor) tuned to 365 nm on a Revolution XD spinning disk confocal microscope. The laser produces 120 μJ pulses at durations of 2–6 ns each. For ablation of cell junctions, 10 consecutive laser pulses were delivered to a single spot along a cell interface. Images were acquired every 4 s before and after ablation of a single interface using a 60x oil immersion lens (Olympus, NA 1.35). The positions of the tricellular vertices connected by the ablated interface were manually tracked prior to and for 36 s following ablation using the image analysis platform SIESTA (*Fernandez-Gonzalez and Zallen, 2011*; *Leung and Fernandez-Gonzalez, 2015*). To measure retraction velocity, the change in distance between the tricellular vertices was measured comparing images acquired immediately before and after ablation and divided by the time elapsed between the two

**Table 2.** Detailed genotypes.

*Figure 1*

| | |
|---|---|
| A | *endo-E-Cadherin::GFP / +; UAS-sh mCherry / matαtub67;15 GAL4* |
| | *endo-E-Cadherin::GFP / +; UAS-sh yrt / matαtub67;15 GAL4* |
| B | *endo-E-Cadherin::GFP / +; UAS-sh mCherry / matαtub67;15 GAL4* |
| | *endo-E-Cadherin::GFP / +; UAS-sh yrt / matαtub67;15 GAL4* |
| C | *endo-E-Cadherin::GFP / UAS-LacZ; daughterless-GAL4 / +* |
| | *endo-E-Cadherin::GFP / +; UAS-FLAG-Yrt / daughterless-GAL4* |
| D | *endo-E-Cadherin::GFP / UAS-LacZ; daughterless-GAL4 / +* |
| | *endo-E-Cadherin::GFP / +; UAS-FLAG-Yrt / daughterless-GAL4* |

*Figure 2*

| | |
|---|---|
| A | *sqh-Sqh::GFP; UAS-sh mCherry / matαtub67;15 GAL4* |
| | *sqh-Sqh::GFP ; UAS-sh yrt / matαtub67;15 GAL4* |
| B | *sqh-Sqh::GFP; UAS-sh mCherry / matαtub67;15 GAL4* |
| | *sqh-Sqh::GFP ; UAS-sh yrt / matαtub67;15 GAL4* |
| C | *sqh-Sqh::GFP; UAS-sh mCherry / matαtub67;15 GAL4* |
| | *sqh-Sqh::GFP ; UAS-sh yrt / matαtub67;15 GAL4* |
| D | *sqh-Sqh::GFP; UAS-sh mCherry / matαtub67;15 GAL4* |
| | *sqh-Sqh::GFP ; UAS-sh yrt / matαtub67;15 GAL4* |
| E | *hs-FLP; sqh-Sqh::GFP; UAS-LacZ / tub-FRT-GAL80-FRT-GAL4, UAS–mRFP* |
| | *hs-FLP; sqh-Sqh::GFP; UAS-FLAG-Yrt / tub-FRT-GAL80-FRT-GAL4, UAS–mRFP* |
| F | *hs-FLP; sqh-Sqh::GFP; UAS-LacZ / tub-FRT-GAL80-FRT-GAL4, UAS–mRFP* |
| | *hs-FLP; sqh-Sqh::GFP; UAS-FLAG-Yrt / tub-FRT-GAL80-FRT-GAL4, UAS–mRFP* |

*Figure 3*

| | |
|---|---|
| A | *Patched-GAL4 / +; UAS-FLAG-GFP / +* |
| | *Patched-GAL4 / +; IAS-FLAG-Yrt / +* |
| B | *Patched-GAL4 / +; UAS-FLAG-GFP / +* |
| | *Patched-GAL4 / +; IAS-FLAG-Yrt / +* |
| C | *hs-FLP / +; ; UAS- LacZ / tub-FRT-GAL80-FRT-GAL4, UAS–mRFP* |
| | *hs-FLP / +; ; UAS- FLAG-Yrt / tub-FRT-GAL80-FRT-GAL4, UAS–mRFP* |
| D | *hs-FLP / +; ; UAS- LacZ / tub-FRT-GAL80-FRT-GAL4, UAS–mRFP* |
| | *hs-FLP / +; ; UAS- FLAG-Yrt / tub-FRT-GAL80-FRT-GAL4, UAS–mRFP* |
| E | *hs-FLP / +; sh sqh / +; UAS- LacZ / tub-FRT-GAL80-FRT-GAL4, UAS–mRFP* |
| | *hs-FLP / +; sh LexA / +; UAS- FLAG-Yrt / tub-FRT-GAL80-FRT-GAL4, UAS–mRFP* |
| | *hs-FLP / +; sh sqh / +; UAS- FLAG-Yrt / tub-FRT-GAL80-FRT-GAL4, UAS–mRFP* |
| F | *hs-FLP / +; sh sqh / +; UAS- LacZ / tub-FRT-GAL80-FRT-GAL4, UAS–mRFP* |
| | *hs-FLP / +; sh LexA / +; UAS- FLAG-Yrt / tub-FRT-GAL80-FRT-GAL4, UAS–mRFP* |
| | *hs-FLP / +; sh sqh / +; UAS- FLAG-Yrt / tub-FRT-GAL80-FRT-GAL4, UAS–mRFP* |

*Figure 4*

| | |
|---|---|
| A | *hs-FLP / +; ; UAS- aPKC[CAAX] / tub-FRT-GAL80-FRT-GAL4, UAS–mRFP* |
| B | *hs-FLP / +; UAS- LacZ / +; UAS- FLAG-Yrt / tub-FRT-GAL80-FRT-GAL4, UAS–mRFP* |
| C | *hs-FLP / +; ; UAS-aPKC[CAAX], UAS- FLAG-Yrt / tub-FRT-GAL80-FRT-GAL4, UAS–mRFP* |
| D | *hs-FLP / +; UAS- LacZ / +; UAS- FLAG-Yrt[5A] / tub-FRT-GAL80-FRT-GAL4, UAS–mRFP* |
| E | *hs-FLP / +; ; UAS-aPKC[CAAX], UAS- FLAG-Yrt[5A] / tub-FRT-GAL80-FRT-GAL4, UAS–mRFP* |
| F | *hs-FLP / +; ; UAS- FLAG-Yrt[5D] / tub-FRT-GAL80-FRT-Gal4, UAS–mRFP* |
| G | *hs-FLP / +; ; UAS- FLAG-Yrt[FWA] / tub-FRT-GAL80-FRT-Gal4, UAS–mRFP* |

*Table 2 continued on next page*

*Table 2 continued*

*Figure 1*

| | |
|---|---|
| H | *hs-FLP / +; ; UAS- aPKC[CAAX] / tub-FRT-GAL80-FRT-GAL4, UAS–mRFP* |
| | *hs-FLP / +; UAS- LacZ / +; UAS- FLAG-Yrt / tub-FRT-GAL80-FRT-GAL4, UAS–mRFP* |
| | *hs-FLP / +; ; UAS-aPKC[CAAX], UAS- FLAG-Yrt / tub-FRT-GAL80-FRT-GAL4, UAS–mRFP* |
| | *hs-FLP / +; UAS- LacZ / +; UAS- FLAG-Yrt[5A] / tub-FRT-GAL80-FRT-GAL4, UAS–mRFP* |
| | *hs-FLP / +; ; UAS-aPKC[CAAX], UAS- FLAG-Yrt[5A] / tub-FRT-GAL80-FRT-GAL4, UAS–mRFP* |
| I | *hs-FLP / +; UAS- LacZ / +; + / tub-FRT-GAL80-FRT-GAL4, UAS–mRFP* |
| | *hs-FLP / +; ; UAS- FLAG-Yrt[5A] / tub-FRT-GAL80-FRT-Gal4, UAS–mRFP* |
| | *hs-FLP / +; ; UAS- FLAG-Yrt[5D] / tub-FRT-GAL80-FRT-Gal4, UAS–mRFP* |
| | *hs-FLP / +; ; UAS- FLAG-Yrt[FWA] / tub-FRT-GAL80-FRT-Gal4, UAS–mRFP* |

*Figure 5*

| | |
|---|---|
| A | *wild type* ovaries treated with DMSO or aPKC inhibitor |
| B | *hs-FLP / +; ; sh aPKC / tub-FRT-GAL80-FRT-GAL4, UAS–mRFP* |
| C | *hs-FLP; tub-GAL4, UAS-GFP / +; FRT82b, crb[11a22] / FRT82b, tub-GAL80* treated with aPKC inhibitor |
| D | *hs-FLP; tub-GAL4, UAS-GFP / +; FRT82b, crb[11a22] / FRT82b, tub-GAL80* treated with aPKC inhibitor |
| E | *hs-FLP; tub-GAL4, UAS-GFP / fosCrb[Y10A]; FRT82b, crb[11a22] / FRT82b,tub-GAL80* treated with aPKC inhibitor |
| F | *hs-FLP; tub-GAL4, UAS-GFP / fosCrb[Y10A]; FRT82b, crb[11a22] / FRT82b,tub-GAL80* treated with aPKC inhibitor |
| G | *hs-FLP; tub-GAL4, UAS-GFP / UAS-LacZ; FRT82b, crb[11a22] / FRT82b,tub-GAL80* |
| | *hs-FLP; tub-GAL4, UAS-GFP / UAS-FLAG-Yrt[5A]; FRT82b / FRT82b,tub-GAL81* |
| | *hs-FLP; tub-GAL4, UAS-GFP / UAS-FLAG-Yrt[5A]; FRT82b, crb[11a22] / FRT82b,tub-GAL80* |
| H | *hs-FLP; tub-GAL4, UAS-GFP / UAS-LacZ; FRT82b, crb[11a22] / FRT82b,tub-GAL80* |
| | *hs-FLP; tub-GAL4, UAS-GFP / UAS-FLAG-Yrt[5A]; FRT82b / FRT82b,tub-GAL81* |
| | *hs-FLP; tub-GAL4, UAS-GFP / UAS-FLAG-Yrt[5A]; FRT82b, crb[11a22] / FRT82b,tub-GAL80* |
| I | *hs-FLP; tub-GAL4, UAS-GFP / UAS-FLAG-Yrt[5A], fosCrb; FRT82b, crb[11a22] / FRT82b,tub-GAL80* |
| | *hs-FLP; tub-GAL4, UAS-GFP / UAS-FLAG-Yrt[5A], fosCrb[Y10A]; FRT82b, crb[11a22] / FRT82b,tub-GAL80* |
| J | *hs-FLP; tub-GAL4, UAS-GFP / UAS-FLAG-Yrt[5A], fosCrb; FRT82b, crb[11a22] / FRT82b,tub-GAL80* |
| | *hs-FLP; tub-GAL4, UAS-GFP / UAS-FLAG-Yrt[5A], fosCrb[Y10A]; FRT82b, crb[11a22] / FRT82b,tub-GAL80* |

*Figure 6*

| | |
|---|---|
| A | hs-FLP / +; ; UAS-sh moe / tub-FRT-GAL80-FRT-GAL4, UAS–mRFP treated with $H_2O$ or aPKC inhibitor |
| B | hs-FLP / +; ; UAS-sh moe / tub-FRT-GAL80-FRT-GAL4, UAS–mRFP treated with $H_2O$ or aPKC inhibitor |
| C | *hs-FLP / +; UAS- myc-Moesin[T559D] / +; + / tub-FRT-GAL80-FRT-GAL4, UAS–mRFP* |
| | *hs-FLP / +; UAS- LacZ / +; UAS- FLAG-Yrt / tub-FRT-GAL80-FRT-GAL4, UAS–mRFP* |
| | *hs-FLP / +; UAS- myc-Moesin[T559D] / +; UAS- FLAG-Yrt / tub-FRT-GAL80-FRT-GAL4, UAS–mRFP* |
| D | *hs-FLP / +; UAS- myc-Moesin[T559D] / +; + / tub-FRT-GAL80-FRT-GAL4, UAS–mRFP* |
| | *hs-FLP / +; UAS- LacZ / +; UAS- FLAG-Yrt / tub-FRT-GAL80-FRT-GAL4, UAS–mRFP* |
| | *hs-FLP / +; UAS- myc-Moesin[T559D] / +; UAS- FLAG-Yrt / tub-FRT-GAL80-FRT-GAL4, UAS–mRFP* |

*Figure 7*

| | |
|---|---|
| A | *hs-FLP / +; ; UAS- LacZ / tub-FRT-Gal80-FRT-Gal4, UAS–mRFP* |
| | *hs-FLP / +; ; UAS-aPKC[CAAX], Par6 / tub-FRT-GAL80-FRT-GAL4, UAS–mRFP* |
| | *hs-FLP / +; ; UAS-Pak[Myr] / +; + / tub-FRT-GAL80-FRT-GAL4, UAS–mRFP* |
| | *hs-FLP / +; ; UAS-Pak[Myr] / +; UAS-aPKC[CAAX], Par6 / tub-FRT-GAL80-FRT-GAL4, UAS–mRFP* |
| B | *hs-FLP / +; ; UAS- LacZ / tub-FRT-Gal80-FRT-Gal4, UAS–mRFP* |
| | *hs-FLP / +; ; UAS-aPKC[CAAX], Par6 / tub-FRT-GAL80-FRT-GAL4, UAS–mRFP* |
| | *hs-FLP / +; ; UAS-Pak[Myr] / +; + / tub-FRT-GAL80-FRT-GAL4, UAS–mRFP* |
| | *hs-FLP / +; ; UAS-Pak[Myr] / +; UAS-aPKC[CAAX], Par6 / tub-FRT-GAL80-FRT-GAL4, UAS–mRFP* |

*Table 2 continued on next page*

*Table 2 continued*

### Figure 1

| | |
|---|---|
| C | *Daughterless-GAL4 / +; UAS-FLAG-GFP / Daughterless-GAL4* |
| | *Daughterless-GAL4 / +; UAS-aPKC[CAAX], Par6 / Daughterless-GAL4* |
| | *Daughterless-GAL4 / UAS-Pak[myr]; + / Daughterless-GAL4* |
| D | *wild type* |
| | *Pp2A-29b[EP2332]* |
| E | *wild type embryos treated with DMSO or PP2A inhibitor* |
| F | *Daughterless-GAL4 / UAS-Pak[myr]; + / Daughterless-GAL4 treated with DMSO or PP2A inhibitor* |

### Figure 8

| | |
|---|---|
| A | *hs-FLP / +; ; UAS- FLAG-Yrt / tub-FRT-GAL80-FRT-GAL4, UAS–mRFP treated with DMSO or Pak1 inhibitor* |
| B | *hs-FLP / +; ; UAS- FLAG-Yrt / tub-FRT-GAL80-FRT-GAL4, UAS–mRFP treated with DMSO or Pak1 inhibitor* |
| C | *hs-FLP / +; sh Pak1; + / tub-FRT-GAL80-FRT-GAL4, UAS–mRFP* |
| | *hs-FLP / +; sh Pak1; UAS- FLAG-Yrt / tub-FRT-GAL80-FRT-GAL4, UAS–mRFP* |
| | *hs-FLP / +; sh Pak1; UAS- FLAG-Yrt[5A] / tub-FRT-GAL80-FRT-GAL4, UAS–mRFP* |
| D | *hs-FLP / +; sh Pak1; + / tub-FRT-GAL80-FRT-GAL4, UAS–mRFP* |
| | *hs-FLP / +; sh lexA; UAS- FLAG-Yrt / tub-FRT-GAL80-FRT-GAL4, UAS–mRFP* |
| | *hs-FLP / +; sh Pak1; UAS- FLAG-Yrt / tub-FRT-GAL80-FRT-GAL4, UAS–mRFP* |
| | *hs-FLP / +; sh lexA; UAS- FLAG-Yrt[5A] / tub-FRT-GAL80-FRT-GAL4, UAS–mRFP* |
| | *hs-FLP / +; sh Pak1; UAS- FLAG-Yrt[5A] / tub-FRT-GAL80-FRT-GAL4, UAS–mRFP* |
| E | *hs-FLP / +; ; UAS- FLAG-Yrt / tub-FRT-GAL80-FRT-GAL4, UAS–mRFP treated with DMSO or PP2A inhibitor* |
| | *hs-FLP / +; ; UAS- FLAG-Yrt[5A] / tub-FRT-GAL80-FRT-GAL4, UAS–mRFP treated with DMSO or PP2A inhibitor* |
| F | *hs-FLP / +; ; UAS- FLAG-Yrt / tub-FRT-GAL80-FRT-GAL4, UAS–mRFP treated with DMSO or PP2A inhibitor* |
| | *hs-FLP / +; ; UAS- FLAG-Yrt[5A] / tub-FRT-GAL80-FRT-GAL4, UAS–mRFP treated with DMSO or PP2A inhibitor* |
| G | *hs-FLP / +; ; sh Pp2A-29b / tub-FRT-GAL80-FRT-GAL4, UAS–mRFP* |
| | *hs-FLP / +; UAS-FLAG-Yrt; sh Pp2A-29b / tub-FRT-GAL80-FRT-GAL4, UAS–mRFP* |
| | *hs-FLP / +; UAS-FLAG-Yrt[5A]; sh Pp2A-29b / tub-FRT-GAL80-FRT-GAL4, UAS–mRFP* |
| H | *hs-FLP / +; ; sh Pp2A-29b / tub-FRT-GAL80-FRT-GAL4, UAS–mRFP* |
| | *hs-FLP / +; sh lexA; UAS- FLAG-Yrt / tub-FRT-GAL80-FRT-GAL4, UAS–mRFP* |
| | *hs-FLP / +; UAS-FLAG-Yrt; sh Pp2A-29b / tub-FRT-GAL80-FRT-GAL4, UAS–mRFP* |
| | *hs-FLP / +; sh lexA; UAS- FLAG-Yrt[5A] / tub-FRT-GAL80-FRT-GAL4, UAS–mRFP* |
| | *hs-FLP / +; UAS-FLAG-Yrt[5A]; sh Pp2A-29b / tub-FRT-GAL80-FRT-GAL4, UAS–mRFP* |

images. To quantify the viscoelastic properties of the tissue, we modeled the laser ablation results as the damped recoil of an elastic fibre, using a Kelvin-Voigt mechanical-equivalent circuit (*Fernandez-Gonzalez et al., 2009*). Briefly, the Kelvin-Voigt circuit models cell interfaces as a spring (elasticity) and a dashpot (viscosity) configured in parallel. As a result, the distance between the ends of the interface when tension is released can be quantified as:

$$L(t) = D\left(1 - e^{-\frac{t}{\tau}}\right)$$

where *L*(*t*) is the distance between the tricellular vertices at the end of the severed interface at time *t* after ablation, *D* is the asymptotic value of the distance between the tricellular junctions, and the relaxation time *τ*:

$$\tau = \frac{\mu}{E}$$

is the ratio of the viscosity ($\mu$) to the elasticity ($E$) of the tissue. Statistical differences between groups were measured using the nonparametric Mann-Whitney U test.

## Immunofluorescence

Dissected ovaries or dechorionated embryos were fixed for 15 min in 4% paraformaldehyde or heat fixed (aPKC, Yrt, Crb, and Arm stainings) as previously described (*Gamblin et al., 2014*). Fixed ovaries and embryos were blocked for 1 hr in NGT (2% normal goat serum, 0.3% Triton X-100 in PBS). Ovaries and embryos were incubated with the following primary antibodies overnight at 4°C: mouse anti-Dlg1 [4F3, Developmental Studies Hybridoma Bank at the University of Iowa (DSHB)], 1:25 dilution; rabbit anti-Yrt, 1:1000 (*Biehler et al., 2020*); guinea-pig anti-Yrt (*Laprise et al., 2006*), 1:250; mouse anti-GFP (Roche, Sigma Aldrich), 1:200; rabbit anti-GFP (A-11122, Invitrogen) 1:200; mouse anti-Fas3 (7G10, DSHB), 1:10; rabbit anti-RFP (600-401-379, Rockland), 1:400; rabbit anti-Lgl (d-300, Santa Cruz), 1:100; rabbit anti-aPKC (C-20, Santa Cruz Biotechnology), 1:200; rat anti-Crb (*Sollier et al., 2015*), 1:250. Ovaries and embryos were then washed three times in PBT (0.3% Triton X-100 in PBS) before and after incubation with secondary antibodies (1:400 in NGT, 1 hr at room temperature), which were conjugated to Cy3 (Jackson ImmunoResearch Laboratories) or Alexa Fluor 488 (Molecular Probes). Ovarioles were mechanically separated and mounted in Vectashield mounting medium (Vector Labs), and imaged with a LSM700 confocal (63× Apochromat lens with a numerical aperture of 1.40). Embryos were mounted in Vectashield mounting medium and imaged with the Zeiss LSM700 confocal using LD C-Apochromat 40x 1.1 NA water Korr objective. Images were uniformly processed with ImageJ (National Institutes of Health).

## Phosphatase assays

Dechorionated embryos were homogenized in ice-cold lysis buffer (1% Triton X-100, 50 mM TRIS-HCl pH 7.5, 5% glycerol, 150 mM NaCl, 1 mM PMSF, 0.5 µg/mL aprotinin, 0.7 µg/mL pepstatin, and 0.5 µg/mL leupeptin). Lysates were cleared by centrifugation at 4°C, and 400 units of λ Phosphatase (New England Biolabs) was added to 30 µg of proteins extracted from embryos. The volume of the reaction mix was completed to 30 µl with the MetalloPhosphatase buffer (New England Biolabs) containing 1 mM of $MnCl_2$ prior to a 30-min incubation at 30°C. The reaction was stopped by addition of Laemmli's buffer.

## Western blotting

Dechorionated embryos were homogenized in ice-cold lysis buffer (1% Triton X-100, 50 mM TRIS HCl pH 7.5, 5% glycerol, 100 mM NaCl, 50 mM NaF, 5 mM EDTA pH 8, 40 mM β-glycerophosphate, 1 mM PMSF, 0.5 µg/mL aprotinin, 0.7 µg/mL pepstatin, 0.5 µg/mL leupeptin and 0.1 mM orthovanadate) and processed for SDS-PAGE and western blotting as previously described (*Laprise et al., 2002*). Primary antibodies used: Rabbit anti-Yrt (*Biehler et al., 2020*), 1:10,000; mouse anti-β−Tubulin (E7, DSHB), 1:2000; rabbit anti-aPKC (C-20, Santa Cruz Biotechnology), 1:2000; rabbit anti-PP2A-A (*Krahn et al., 2009*), 1:10,000; mouse anti-Actin (NB-100–74340, Novus Biologicals), 1:2000. HRP-conjugated secondary antibodies were from GE Healthcare and used at a 1:2000 dilution.

## Chemical treatment of embryos

Dechorionated embryos were incubated under agitation for 1 hr at room temperature in a 0.9% NaCl solution supplemented with 100 µM of Cantharidin under an octane phase (1:1). Embryos were then washed three times in PBT.

## Chemical treatment of ovaries

Dissected ovaries were treated for 2 hr with inhibitors of aPKC (CRT-006-68-54, Bio-Techn Sales Corporation, 10 µM) or Pak1 (IPA-3, Millipore Sigma, 50 µM) as described (*Chartier et al., 2012*).

## Measurements of apical domain size and quantification of fluorescence intensity

Stage 3-6 follicles were imaged, and the apical diameter or fluorescence intensity of genetically modified cells (positively labeled with mRFP or GFP) was measured using ImageJ (National Institutes of Health). Control cells facing labeled cells on the opposite side of the follicle were used as control.

Results are expressed as the ratio between measurements in labeled cells and control cells. A minimum of 15 follicles were analyzed in each condition. A similar approach was used to analyse apical domain width of epidermal cells expressing FLAG-GFP (control) or FLAG-Yrt in stage 13 embryos (n $\geq$ 13 from 7 different embryos). FLAG-GFP- or FLAG-Yrt-negative cells in the same embryonic segment were used as control. Statistical differences between groups were determined using one-way ANOVA (apical domain size) or Student's t-test (fluorescence intensity).

### Analysis of myosin distribution in the embryonic epidermis

To analyze myosin distribution in epidermal cells in stage 13 and stage 14 embryos, image stacks were acquired of embryos expressing sqh:GFP and either shRNA targeting *mCherry* or *yrt*. Maximum intensity projections were generated from slices representing the apical domain (5 slices of 0.5 µm each) of the epidermis. A 20 µm line was drawn along the anterior-posterior axis of the tissue in each embryo to calculate mean intensity and heterogeneity (*Zulueta-Coarasa and Fernandez-Gonzalez, 2018*) of the GFP signal. Here, heterogeneity is defined as:

$$\frac{standard\,deviation(intensity)}{mean(intensity)}$$

This measurement captures the variability within each linescan, which increases when there are clear peaks and troughs.

## Acknowledgements

The authors would like to acknowledge A Wodarz, S Campuzano, T Harris, E Knust, D St-Johston, Y Bellaiche, N Perrimon, the Bloomington *Drosophila* Stock Center, the *Drosophila* Genomics Resource Center, and the Developmental Studies Hybridoma Bank for reagents. Flybase was used as an important database for this work. We also thank C Gamblin for critical reading of the manuscript. This work was supported by operating grants from the Canadian Institute of Health Research (CIHR) to PL (MOP-142236) and RFG (MOP-156279). CB is supported by a scholarship from FRQ-S, and K R is a postdoctoral CIHR grantee.

## Additional information

### Funding

| Funder | Grant reference number | Author |
| --- | --- | --- |
| Canadian Institutes of Health Research | MOP-142236 | Patrick Laprise |
| Canadian Institutes of Health Research | MOP-156279 | Rodrigo Fernandez-Gonzalez |
| Canadian Institutes of Health Research | | Katheryn E Rothenberg |
| Fonds de Recherche du Québec - Santé | | Cornelia Biehler |

The funders had no role in study design, data collection and interpretation, or the decision to submit the work for publication.

### Author contributions

Cornelia Biehler, Conceptualization, Data curation, Formal analysis, Investigation, Methodology, Writing - review and editing; Katheryn E Rothenberg, Data curation, Formal analysis, Investigation, Writing - review and editing; Alexandra Jette, Formal analysis, Methodology, Writing - review and editing; Helori-Mael Gaude, Data curation, Investigation; Rodrigo Fernandez-Gonzalez, Conceptualization, Formal analysis, Supervision, Funding acquisition, Investigation, Writing - review and editing; Patrick Laprise, Conceptualization, Resources, Formal analysis, Supervision, Funding acquisition, Writing - original draft, Project administration, Writing - review and editing

## Author ORCIDs

Katheryn E Rothenberg http://orcid.org/0000-0002-8191-7528
Alexandra Jette http://orcid.org/0000-0001-5423-2374
Rodrigo Fernandez-Gonzalez http://orcid.org/0000-0003-0770-744X
Patrick Laprise https://orcid.org/0000-0001-9785-4376

## Decision letter and Author response

Decision letter https://doi.org/10.7554/eLife.67999.sa1
Author response https://doi.org/10.7554/eLife.67999.sa2

# Additional files

## Supplementary files

• Transparent reporting form

## Data availability

Source data have been provided for all figures and are available as supporting files.

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
