## [Decision Letter]

[Editors' note: this paper was reviewed by Review Commons.]

**Acceptance summary:**

In this manuscript, the authors investigated how the balance of apical and lateral membrane domains is regulated in epithelial cells of the *Drosophila* embryonic epidermis. They found that the protein Yurt, which controls the function of the apical determinant Crumbs, is regulated by a balance of phosphorylation/de-phosphorylation by PP2A and aPKC. Their results revealed that these interactions reciprocally regulate apical contractility and polarization, thereby providing a homeostatic mechanism that may link cell polarity and tissue mechanics.

---

## [Author Response]

Reviewer #1 (Evidence, reproducibility and clarity (Required)):Atypical protein kinase C (aPKC) and p21-activated kinase 1 (Pak1) are kinases acting downstream of Cdc42 that have many common substrates and redundant activity on target proteins and epithelial cell polarity. In this work, authors find that they have opposing effects on Yurt. Unphosphorylated Yurt is recruited to the apical membrane by Crumbs and increases Myosin dependent apical tension. While aPKC-dependent phosphorylation of Yurt represses this apical recruitment, Pak1 mediated activation of Protein phosphatase 2A dephosphorylates Yurt thereby increasing apical localization and tension.This is an interesting manuscript that presents novel insights into how cortical tension is regulated by Yurt. The experimental data broadly support the conclusions and the discovery that aPKC and Pak1 have opposite effects on Yurt is a welcome observation in light of the recent claim that these kinases are redundant with each other. The manuscript could be improved in several ways, but the first two are not essential for publication:1. All of the experiments involve artificial conditions in which Yurt is over-expressed. However, aPKC is active in most if not all epithelia and Yurt is therefore restricted to the lateral domain. Is there any evidence that changes in the apical recruitment of Yurt modify cortical tension in wild-type cells during development? Demonstrating that the apical localisation of Yurt is developmentally regulated and plays a role in morphogenesis would substantially increase the significance of their results.

We fully agree with reviewer 1 that demonstrating the importance of the pathway that we described in a physiological process during development would greatly increase the significance of our study. We have previously showed that the apical recruitment of Yrt is developmentally regulated (Laprise et al., 2006). Yrt is confined to the lateral membrane in embryonic epithelial tissues from the end of cellularization till stage 13. At stage 14, Yrt is recruited apically by Crb and localizes to both apical and lateral membranes in the epidermis. Specifically, Yrt is enriched at lateral septate junctions, and also co-localizes with Crb within the marginal zone (Laprise et al., 2006; Laprise et al., 2009). We found that knockdown of *yrt* decreased cortical Sqh levels at the apex of epidermal cells from stage 14 of embryogenesis, whereas Yrt depletion had no major impact of Myosin distribution at earlier stages of development (Figure 2 of the revised manuscript). This strongly argues that the apical recruitment of Yrt confers to the latter the ability to control Myosin-dependent function. In addition, Yrt-depleted cells showed delayed elongation along the dorso-ventral axis at stage 14. This suggests that the Yrt-dependent recruitment of Myosin contributes to cell elongation along the dorsal-ventral axis in preparation for dorsal closure, thereby providing a putative molecular and cellular basis explaining defective dorsal closure in *yrt* mutant embryos (Hoover and Bryant, 2002; Laprise et al., 2006).

2. Along similar lines, does unphosphorylated Yurt increase cortical tension laterally when localised there or does the recruitment of Myosin also require Crumbs?

Yrt only impacts Myosin distribution following its apical recruitment in the embryonic epidermis. Morever, Yrt overexpression recruits Myosin specifically at the apical membrane (Figure 2E, F) in follicle cells. This recruitment requires the apical protein Crb (Figure Author response image 1). Consistent with these results, overexpression of wild type Yrt or nonphosphorylatable Yrt^5A^ decreases cell diameter exclusively at the cell apex, a function that requires Crb (Figure 3A-D; 5G, H). In addition, mutation of the FERM-binding domain of Crb, which is required for apical recruitment of Yrt (Laprise et al., 2006), abolished Yrtinduced apical constriction (Figure 5I, J). We thus believe that Yrt primarily acts downstream of Crb to increase contractility at the apical domain, and that Yrt has a limited role in promoting cortical tension at the lateral domain. Our data thus propose that Crb-bounded Yrt promotes cortical tension (this study), whereas lateral Yrt maintains lateral membrane identity and controls septate junction permeability [Figure 9 of the revised manuscript; (Laprise et al., 2006; Laprise et al., 2009)].

**Author response image 1. respfig1:** Yrt is a multifunctional protein promoting cortical tension downstream of Crb. Mosaic expression of FLAG-Yrt (F-Yrt) with a control shRNA or a shRNA targeting *crb* in follicular epithelial cells constitutively expressing Sqh::GFP, which was visualized by immunofluorescence. FLAG-Yrt-expressing cells (middle and right panels) are labeled with mRFP. The lower part of the figure shows quantification of apical Sqh intensity in cells expressing the transgenes listed above. Results are presented as a ratio between clonal cells and control cells in the same follicle (see Author response image 2; **P* £ 0.05, ***P* £ 0.01).

3. Most of the images are tiny and show only small regions of the embryo or follicular epithelium, which makes it very difficult to tell which part of the tissue these come from. Cortical tension varies with position and developmental stage in the follicle cell layer (see for example Balaji et al. (2019) Development: 146, dev171256), and it is therefore important to specify both, either by showing a larger region of each egg chamber or by putting this information in the Figure legends. To illustrate this point, figure 2J is meant to show that cells treated with the aPKC inhibitor have a reduced apical area, but the righthand image looks like wild-type cells at either the anterior or posterior end of the egg chamber, where the epithelium is more curved.

We thank reviewer 1 for mentioning the work of Balaji et al. Of note, we only used previtellogenic follicles for quantification (stage 3-6 follicles; this information is available in the paper, see p.20). At these stages, follicles display limited elongation, and all follicular epithelial cells are cuboidal. This suggests a relatively homogenous tension across the tissue. Accordingly, Balaji et al. showed that differential Myosin distribution and the ensuing modulation of tension in follicular epithelial cells mostly occur between stage 6 to 9 when egg chambers lengthen, and epithelial cells acquire different morphologies along the antero-posterior axis. In addition, we haven’t observed variation in phenotypes related to cell position upon Yrt overexpression. However, we recognize that cortical tension may vary with position within the follicle, even at pre-vitellogenic stages. This is why we always selected cells facing mutant/overexpressing cell clones in the follicle as control (mirror image). In this way, control cells are at the same developmental stage and occupy a position within the follicle with similar physical constrains. This is explained in Methods and illustrated in Figure Author response image 2. In parallel to acquiring high magnification images, which facilitate visualization of apical constriction in the paper, we systematically imaged the entire follicle (or a large portion of it) for quantification of apical domain width of control and mutant/transgene-expressing cells. This enabled us to provide the stage of all follicles in figure legends of the revised manuscript, as requested by reviewer 1.

**Author response image 2. respfig2:** We used mosaic analysis in the *Drosophila* follicular epithelium. The impact of clonal gene depletion and/or protein overexpression on apical constriction was assessed by measuring apical domain width of cells within clones (pink cells) and control cells occupying an equivalent position on the other side of the follicle (grey cells). Results were expressed as the ratio between the width of clonal cells and the width of control cells in the same follicle.

4. Although the panels present examples that prove the relevant point for the specific figure, there are some inconsistencies across figures. For example, Yurt localises apically in Figure 2K, but there is no obvious apical constriction.

It is true that inhibition or knockdown of aPKC (shown as Figure 5A, B in the revised manuscript) is not sufficient to significantly induce apical constriction although Yrt is partially relocalized apically under these conditions. As explained in the discussion (see pages 12 and 13 of the manuscript), our model is that Yrt competes with other FERM domain proteins, such as Moesin (Moe), for binding to Crb. Of note, Yrt and Moe have opposite effects on Myosin activity downstream of Crb [this study; (Flores-Benitez and Knust, 2015; Laprise et al., 2006; Medina et al., 2002; Salis et al., 2017)]. It is thus possible that the amount of Yrt relocalized to the apical domain upon inhibition of aPKC is not sufficient to fully outcompete Moe, whereas Yrt overexpression reaches the threshold required to displace most Moesin molecules. Accordingly, we found that inhibition of aPKC in *Moe* knockdown cells caused apical constriction, in contrast to what was observed in control cells (Figure 6A, B of the revised manuscript). In addition, we found that expression of active Moe suppresses Yrt-induced apical constriction (Figure 6C, D), thereby supporting the concept that these proteins are involved in a competitive functional interaction. These data provide further support to our model that Crb functions are specified by FERM domain proteins that competitively associate with Crb to form distinct complexes.

Specific comments:1. Figure 2L/2O : It is not clear if Yurt is relocalizing in 2L/O in the control cells on aPKC inhibitor treatment. Quantification along with a better image would help readers appreciate this change.

We provided better images along with quantification confirming that the apical relocalization of Yrt upon inhibition of aPKC requires Crb with an intact FERM-domain binding motif (Figure 5C-F of the revised manuscript).

2. Figure 3B – Top right: The Yrt/ aPKC signal is not clear. It would help to provide a better image.

We replaced Figure 3B (now 7A in the revised manuscript), which confirms that expression of aPKC^CAAX^ together with Par6 reduces membrane association of Yrt. This has also been quantified (see Figure 7B of the revised manuscript).

3) The evidence that PAK1 regulates Yrt phosphorylation rests almost entirely on the treatment with a Pak1 inhibitor, as the expression of myristolated Pak1 has little effect on Yurt phosphorylation (Figure 3C). Since inhibitors often have off target effects, it would help to confirm this results by making pak1 mutant clones.

We showed that shRNA-mediated knockdown of *Pak1* suppresses apical constriction induced by Yrt overexpression (Figure 8C, D of the revised manuscript). This complements the use of the Pak1 inhibitor (Figure 8A, B) and demonstrates that Pak1 is required for Yrtdependent apical constriction. To supplement Figure 3C (now Figure 7C) and further support our conclusion that Pak1 sustains Yrt function by reducing its phosphorylation on aPKC target sites, we expressed Yrt^5A^ with a control shRNA or a shRNA targeting *Pak1* (the S/T residues targeted by aPKC are mutated to non-phosphorylatable A residues in Yrt^5A^). We found that Yrt^5A^ induces apical constriction in Pak1-depleted cells whereas wild type Yrt is unable to do so in absence of Pak1. This solidifies our model indicating that Pak1 support Yrt function by preventing phosphorylation of aPKC target sites. This result was added as Figure 8C-D of the revised manuscript.

Reviewer #1 (Significance (Required)):This is an interesting manuscript that presents novel insights into how cortical tension is regulated by Yurt. The experimental data broadly support the conclusions and the discovery that aPKC and Pak1 have opposite effects on Yurt is a welcome observation in light of the recent claim that these kinases are redundant with each other.All of the experiments involve artificial conditions in which Yurt is over-expressed. However, aPKC is active in most if not all epithelia and Yurt is therefore restricted to the lateral domain. Is there any evidence that changes in the apical recruitment of Yurt modify cortical tension in wild-type cells during development? Demonstrating that the apical localisation of Yurt is developmentally regulated and plays a role in morphogenesis would substantially increase the significance of their results.Along similar lines, does unphosphorylated Yurt increase cortical tension laterally when localised there or does the recruitment of Myosin also require Crumbs?Reviewer #2 (Evidence, reproducibility and clarity (Required)):SummaryThe manuscript by Biehler et al. addresses the role and regulation of Yrt in promoting cortical tension and apical constriction. The authors show using laser ablation that Yrt is necessary and sufficient to promote cortical tension. Moreover, Yrt overexpression results in accumulation of Myosin and apical cell constriction. Furthermore, Yrt's ability to promote apical constriction depends on Crb and is counteracted by the kinase aPKC. Finally, overexpression of a membrane-targeted form of the kinase Pak1 decreases Yrt phosphorylation in a PP2A (a phosphatase) dependent manner. PP2A antagonizes aPKC's function on Yrt. The authors conclude that Pak1-induced PP2A activity antagonizes aPKC to promote Yrt-induced apical constriction.Major comments1. Figure 1 A-D It is unclear why the measured retraction velocities of the two controls in B and D are so different. The difference in mean retraction velocity between the two controls exceeds the differences in mean retraction velocity between controls and experiments. The authors should comment (or repeat the laser ablation experiments). The authors also should show the retraction as a function of time after ablation.

The differences in retraction velocities in our original data set could be explained by the large gap in time (~6 months) between the experiments, which could have affected laser power and other experimental conditions. We repeated all the laser ablation experiments within 2 weeks and replaced the data in Figure 1. There is now no significant difference between the controls in Figures 1B and 1D. As before, *yrt* knockdown resulted in slower retraction velocities (reduced cortical tension), and Yrt overexpression increased retraction velocities (increased tension). We included kymographs of the junction ablation experiments in Figures 1A and 1C to show retraction over time.

2. Figure 1E,F It is unclear how yrt-dependent changes in viscosity relate to alterations in cortical tension. The authors should clarify.

We updated our calculations of viscoelasticity of the tissue and found that there was no significant effect of Yrt manipulation on tissue viscosity. Previously, we fitted the individual retraction vs. time curves with the Kelvin-Voigt model out to 68 seconds following ablation. However, we realized that the goodness of fit decreased significantly when we included data beyond 40 seconds after laser ablation (Author response image 3, left). The reason for this is that the Kelvin-Voigt model does not fit well the stress relaxation that occurs at longer time scales. Thus, we fit the initial 36 seconds post-ablation of the newly acquired data, finding that the goodness of fit provides R values (correlation between data and model) consistently greater than 0.96 (Author response image 3, right). With this improved calculation, we found no effect of Yrt manipulation on relaxation time, and thus no effect on tissue viscosity: relaxation times were 12 ± 1 seconds for *yrt* knockdown vs. 11 ± 2 seconds for controls, and 8 ± 1 seconds for Yrt overexpression vs 10 ± 1 seconds for controls. Thus, changes in recoil velocity can be attributed to changes in cortical tension.

**Author response image 3. sa2fig3:** Kelvin-Voigt model fits retraction over time better at shorter time scales. (Left) A sample set of retraction vs. time data (points) fit with the Kelvin-Voigt model (line). The accuracy of the fit for the initial recoil is sacrificed in an attempt to fit the longer stress relaxation response. (Right) The same data fit with the Kelvin-Voigt model only up to 36 seconds post-ablation (dashed line). The fit for the initial recoil response is significantly more accurate.

3. Figure 3A The use of a PAK1 inhibitor is not convincing. The authors should use pak1 mutant clones (or clones expressing sh-RNA targeting Pak1 ,see Figure 4) that allow a direct comparison of Yrb subcellular distribution in control and pak1 mutant cells.

We found that shRNA-mediated knockdown of *Pak1* and inhibition of Pak1 activity with the chemical inhibitor IPA-3 both suppressed Yrt-induced apical constriction (Figure 8A-D of the revised manuscript). Combination of these complementary approaches confirms that Pak1 is required to sustain Yrt function, thereby supporting our model. In the original manuscript, we also showed that IPA-3 releases Yrt from the lateral membrane. We were not able to convincingly reproduce this phenotype by knocking down *Pak1*. Our hypothesis is that residual Pak1 in *Pak1*-knockdown cells is sufficient to maintain some Yrt at the lateral membrane. However, we cannot exclude that IPA-3 may have additional targets impacting Yrt localization as suggested by reviewer 2. We thus decided to remove this result from the manuscript to avoid any possible confusion. Importantly, we showed that expression of active Pak1^Myr^ suppresses aPKC-induced displacement of Yrt from the membrane (see Figure 7B of the revised manuscript for quantification), further supporting our model that Pak1 and aPKC have antagonistic roles in regulating Yrt localization and function. Moreover, our data indicate that Yrt^5A^ induces apical constriction in Pak1-depleted cells whereas wild type Yrt is unable to do so in absence of Pak1 (Figure 8C-D of the revised manuscript; the S/T residues targeted by aPKC are mutated to non-phosphorylatable A residues in Yrt^5A^). This indicates that Pak1 is dispensable for Yrt-induced apical constriction when aPKC target sites on Yrt are mutated to non phosphorylatable A residues. Altogether, our data indicates that Pak1 supports Yrt function by preventing phosphorylation of aPKC target sites.

Minor comments4. Title: The title appears to be too general. A more informative title reflecting the interplay between Yrb/aPKC/Pak1 should be chosen.

The new title is: Pak1 and PP2A antagonize aPKC function to support cortical tension induced by the Crumbs-Yurt complex.

5. Abstract: "…we show that Yurt is not a general inhibitor of crumbs…" This should be rephrased as the authors only address one specific function of Crb in their manuscript.

The sentence now reads as follow: ‘Here, we show that Yurt also increases Myosin dependent cortical tension downstream of Crumbs. Yurt overexpression thus induces apical constriction in epithelial cells.’

6. Figure 1 The authors should clarify why they measured tension in one epithelium (embryonic epidermis) and did the remaining analyses in another epithelium (follicle epithelium).

We measured tension in the embryonic epidermis because this is a well-established model for such experiments, and our experimental setup is adapted to this tissue. The apical surface of the embryonic epidermis faces the exterior of the embryo. In contrast, the apical surface of the follicular epithelium faces the interior of the egg chamber. Using the embryonic epidermis for laser ablation assays facilitates laser focusing on apical and subapical junctional regions, minimizing the loss of laser power caused by light travelling through cells. The laser power lost would be significantly greater if we targeted (sub)apical regions in the follicular epithelium.

We then moved to the follicular epithelium to monitor apical constriction, as cell organization in this tissue is well suited for visualization of the apical-basal axis. In addition, defects in tissue architecture are much less pronounced in the follicular epithelium upon alteration of epithelial polarity or cell contractility, thereby facilitating the analysis of cell shape and integrity. Additionally, for genetic manipulations, the follicular epithelium provides a much more tractable experimental system in which clonal analysis is more effective than in the embryo. However, we have data showing that Yrt also induces apical constriction in the embryonic epidermis similar to what we observed in the follicular epithelium, showing that Yrt promotes apical constriction in both embryonic and adult tissues. This result was added to the manuscript as Figure 3A, B.

7. Page 8, top "…demonstrate that the Crb-YRT complex promotes cortical tension and apical constriction." The authors did not measure cortical tension in this context. The authors should rephrase their conclusion more carefully.

We rephrased this part of the manuscript, which is now: ‘Together, these results demonstrate that the Crb–Yrt complex promotes apical constriction’.

8. Figure 3C should indicate aPKC-CAAX + Par6 and not only aPKC-CAAX.

This has been fixed.

9. It would be helpful for the reader if the authors could provide a summary/model of their work as an additional panel to Figure 4.

We added a schematic model summarizing our findings (Figure 9 in the manuscript).

10. Discussion: The data referred to as 'unpublished results' should be shown in supplementary figures.

We think that showing these results would break the flow of the paper. We simply removed the part of the discussion referring to these unpublished data, which are not necessary to support our conclusions. In addition, a published paper referenced in the discussion shows similar results (Salis et al., 2017). This also shortens the discussion, as requested by reviewer 2 (see 11, below).

11. Discussion: The discussion is long and should be written more concisely. The mere repetition of the data already shown in the results part should be minimized.

We trimmed the discussion and removed unnecessary/repetitive parts.

Reviewer #2 (Significance (Required)):SignificanceThe manuscript provides a conceptual advance in the understanding of how interactions between regulatory proteins control mechanical tension and cell shape in *Drosophila* epithelia. Previous work showed that (i) Yrt is essential for epithelial polarity (Laprise, 2009), (ii) Yrt subapical localization depends on Crb (Salis, 2017), (iii) Yrt function is antagonized by aPKC (Gamblin, 2018; Gamblin, 2014) and (iv) Yrt promotes apical cell constriction in the pupal wing (Salis, 2017). The manuscript now adds two players (Pak1, PP2A) to the regulation of Yrt activity and clarifies Yrt's role in epithelial morphogenesis.ReadershipThe manuscript might be of interest to researchers studying cell polarity and epithelial morphogenesis in animals.Field of expertise*Drosophila*Reviewer #3 (Evidence, reproducibility and clarity (Required)):In this manuscript Biehler and co-workers analyzed the role of Yurt (Yrt) a protein know to anatgonize the role of Crumbs (Crb) in apical membrane growth. Using *Drosophila* (embryos and egg chambers) as a model system they found that Yrt promotes myosinIImediated apical constriction and that aPKC prevents Crb-Yrt interaction through direct phosphorylation of Yrt, which results in Yrt dissociation from the apical surface and repression of apical constriction. In contrast, the kinase Pak1 promotes Yrt dephosphorylation, through activation of the phosphatase PP2A, facilitating apical constriction. In summary this manuscript provides evidence for a role for Yrt in apical constriction but it remains unclear how this is related to the role of Crumbs in limiting apical membrane growth.Comments:1) The snapshot of the laser ablation experiments presented in Figure 1 demonstrate the presence of large fluorescence aggregate after photoablation (panel A and C). This suggests that rather that punctual cutting of junctional structures the laser light might have caused the production of cavitation bubbles. No videos for these experiments have been provided making it difficult to evaluate the validity of these perturbations. Also, it is unclear how many times these experiments have been repeated, n=17 junctions, does this refer to 17 junctions in the same embryo? If so, it must be repeated at least in three different embryos.

The circular marks that are visible following laser ablation are not cavitation bubbles, but auto fluorescent holes made by the laser in the vitelline membrane, the protective sac that encloses the *Drosophila* embryo. To show this, we have attached both a maximum intensity projection of the tissue (the approach used to generate the images shown in Figure 1A and 1C) and the individual Z-slices that make up the projected image in an embryo expressing DE-cadherin::GFP (Author response image 4). The first slice labelled Z = 0 µm is the plane containing the vitelline membrane. The vitelline membrane plane displays weak autofluorescence and the bright-rimmed hole caused by laser ablation. The outline of the hole is bright due to ultraviolet-induced photoactivation of the vitelline membrane. Subsequent slices move basally into the epidermis. The slice at Z = 0.5 µm is the plane in which the ablation laser was focused, corresponding to the plane of adherens junctions. Because the surface of the embryo is curved, travelling further in Z brings adherens junctions further from the center of the image into focus. When the maximum intensity projection is generated, the image shows the circular ring apparently in the plane of the severed junction, but this is just a result of the projection. Note that over time (Video 1 and 2), the intensity of the mark on the vitelline membrane decays, consistent with it being the result of UV-induced photoactivation. In addition, the fluorescent mark on the vitelline membrane does not move despite movements of the cells (Video 1, 2), further confirming that the fluorescent mark is in a static structure (the vitelline membrane is glued to the coverslip). We offer a brief explanation of the circular fluorescent marks in the Figure 1 legend.

**Author response image 4. sa2fig4:** Autofluorescent holes in the vitelline membrane generated by laser ablation. The leftmost image shows the maximum intensity projection of a section of epidermal tissue immediately following ablation. The rest of the images are individual confocal slices starting at the plane of the vitelline membrane (Z = 0 um) where the hole caused by ablation is in focus, and moving basally into the tissue. Note that the ablated adherens junction is in focus at the plane Z = 0.5 um.

We have also included videos of the laser ablation experiments as supplemental material.

The number of experiments reported corresponds to the number of junctions severed. Only one junction was ablated in any given embryo. Therefore, *n* = 17 junctions indicates that 17 embryos were imaged with one junction ablated in each embryo. We have adjusted the figure legend to clarify this point.

2. Experiments have been performed in two different model epithelia, however there is no explanation of why some experiments were done in one system and some in the other.

Please refer to comment no. 6 of reviewer 2 for a detailed answer to this point.

3. The authors should specify for each of the panel presented in how many embryos/egg chambers were the experiments repeated. It seems all the quantifications were made using multiple cells form one sample.

We used multiple independent samples. This was clarified in both methods and figure legends.

Reviewer #3 (Significance (Required)):The results presented in this stud expand our knowledge on how polarity proteins regulate epithelial cell polarization and cell contractility but are overall rather incremental compared to what it is known already, which limits my overall enthusiasm.Referees cross-commentingIn light of reviewer 2 and my comments, I think it will be particularly important that the authors clarify the laser ablation experiments as detailed in the report.

References

Flores-Benitez, D., and E. Knust. 2015. Crumbs is an essential regulator of cytoskeletal dynamics and cell-cell adhesion during dorsal closure in *Drosophila*. *eLife*. 4.

Hoover, K.B., and P.J. Bryant. 2002. *Drosophila* Yurt is a new protein-4.1-like protein required for epithelial morphogenesis. *Dev Genes Evol*. 212:230-238.

Karagiosis, S.A., and D.F. Ready. 2004. Moesin contributes an essential structural role in *Drosophila* photoreceptor morphogenesis. *Development*. 131:725-732.

Laprise, P., S. Beronja, N.F. Silva-Gagliardi, M. Pellikka, A.M. Jensen, C.J. McGlade, and U. Tepass. 2006. The FERM protein Yurt is a negative regulatory component of the Crumbs complex that controls epithelial polarity and apical membrane size. *Developmental cell*. 11:363-374.

Laprise, P., K.M. Lau, K.P. Harris, N.F. Silva-Gagliardi, S.M. Paul, S. Beronja, G.J. Beitel, C.J. McGlade, and U. Tepass. 2009. Yurt, Coracle, Neurexin IV and the Na(+),K(+)-ATPase form a novel group of epithelial polarity proteins. *Nature*. 459:1141-1145.

Laprise, P., and U. Tepass. 2011. Novel insights into epithelial polarity proteins in *Drosophila*. *Trends Cell Biol*.

Medina, E., J. Williams, E. Klipfell, D. Zarnescu, G. Thomas, and A. Le Bivic. 2002. Crumbs interacts with moesin and β(Heavy)-spectrin in the apical membrane skeleton of *Drosophila*. *The Journal of cell biology*. 158:941-951.

Salis, P., F. Payre, P. Valenti, E. Bazellieres, A. Le Bivic, and G. Mottola. 2017. Crumbs, Moesin and Yurt regulate junctional stability and dynamics for a proper morphogenesis of the *Drosophila* pupal wing epithelium. *Sci Rep*. 7:16778.

Sollier, K., H.M. Gaude, F.J. Chartier, and P. Laprise. 2015. Rac1 controls epithelial tube length through the apical secretion and polarity pathways. *Biol Open*. 5:49-54.

Wodarz, A., U. Hinz, M. Engelbert, and E. Knust. 1995. Expression of crumbs confers apical character on plasma membrane domains of ectodermal epithelia of *Drosophila*. *Cell*. 82:67-76.